# Should We Learn
# Most Likely Functions or Parameters?

**Shikai Qiu**[*]  **Tim G. J. Rudner**[*]  **Sanyam Kapoor**[*]  **Andrew Gordon Wilson**

New York University

## Abstract

Standard regularized training procedures correspond to maximizing a posterior distribution over parameters, known as maximum a posteriori (MAP) estimation. However, model parameters are of interest only insomuch as they combine with the functional form of a model to provide a function that can make good predictions. Moreover, the most likely parameters under the parameter posterior do not generally correspond to the most likely function induced by the parameter posterior. In fact, we can re-parametrize a model such that any setting of parameters can maximize the parameter posterior. As an alternative, we investigate the benefits and drawbacks of directly estimating the most likely function implied by the model and the data. We show that this procedure leads to pathological solutions when using neural networks and prove conditions under which the procedure is well-behaved, as well as a scalable approximation. Under these conditions, we find that function-space MAP estimation can lead to flatter minima, better generalization, and improved robustness to overfitting.

## 1   Introduction

Machine learning has matured to the point where we often take key design decisions for granted. One of the most fundamental such decisions is the loss function we use to train our models. Minimizing standard regularized loss functions, including cross-entropy for classification and mean-squared or mean-absolute error for regression, with $\ell_1$ or $\ell_2$ regularization, exactly corresponds to maximizing a posterior distribution over model parameters [2, 19]. This standard procedure is known in probabilistic modeling as *maximum a posteriori* (MAP) parameter estimation. However, parameters have no meaning independent of the functional form of the models they parameterize. In particular, our models $f_\theta(x)$ are functions given parameters $\theta$, which map inputs $x$ (e.g., images, spatial locations, etc.) to targets (e.g., softmax probabilities, regression outputs, etc.). We are typically only directly interested in the function and its properties, such as smoothness, which we use to make predictions.

Alarmingly, the function corresponding to the most likely parameters under the parameter posterior does not generally correspond to the most likely function under the function posterior. For example, in Figure 1a, we visualize the posterior over a mixture coefficient $\theta_R$ in a Gaussian mixture regression model in both parameter and function space, using the same Gaussian prior for the mixture coefficients (corresponding to $\ell_2$ regularization in parameter space). We see that each distribution is maximized by a different parameter $\theta_R$, leading to very different learned functions in Figure 1b. Moreover, we can re-parametrize the functional form of any model such that any arbitrary setting of parameters maximizes the parameter posterior (we provide further discussion and an example in Appendix A).

*Should we then be learning the most likely functions or parameters?* As we will see, the nuanced pros and cons of each approach are fascinating and often unexpected.

---

[*]Equal contribution.

37th Conference on Neural Information Processing Systems (NeurIPS 2023).

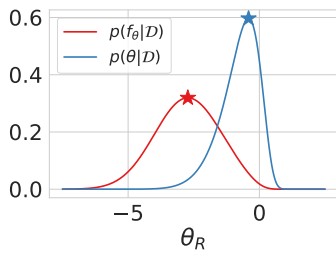

(a) Distinct Posterior Densities

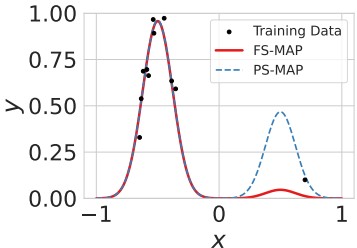

(b) Distinct Learned Functions

Figure 1: **Illustration of the Difference Between Most Likely Functions and Parameters.** The function that is most probable (denoted as FS-MAP) under the posterior distribution over functions can substantially diverge from the function represented by the most probable parameters (denoted as PS-MAP) under the posterior distribution over parameters. We illustrate this fact with a regression model with two parameters, $\theta_L$ and $\theta_R$, both with a prior $\mathcal{N}(0, 1.2^2)$. This model is used to learn a mixture of two Gaussians with fixed mean and variance, where the mixture coefficients are given by $\exp(\theta_L)$ and $\exp(\theta_R)$. Both the most probable solution and the posterior density show significant differences when analyzed in function-space versus parameter-space. Since $\theta_L$ is well-determined, we only plot the posterior as a function of $\theta_R$. We normalize the area under $p(f_\theta|\mathcal{D})$ to 1.

On one hand, we might expect the answer to be a clear-cut "We should learn most likely functions!". In Section 3, we present a well-defined function-space MAP objective through a generalization of the change of variables formula for probability densities. In addition to functions being the direct quantity of interest and the function-space MAP objective being invariant to the parametrization of our model, we show that optimization of the function-space MAP objective indeed results in more probable functions than standard parameter-space MAP, and these functions often correspond to *flat minima* [10, 12] that are more robust to overfitting.

On the other hand, function-space MAP is not without its own pathologies and practical limitations. We show in Section 4 that the Jacobian term in the function-space MAP objective admits trivial solutions with infinite posterior density and can require orders of magnitude more computation and memory than parameter-space MAP, making it difficult to scale to modern neural networks. We also show that function-space MAP will not necessarily be closer than parameter-space MAP to the posterior expectation over functions, which in Bayesian inference forms the mean of the posterior predictive distribution and has desirable generalization properties [2, 19, 29]. To help address the computational limitations, we provide in Section 4 a scalable approximation to the function-space MAP objective applicable to large neural networks using Laplacian regularization, which we refer to as L-MAP. We show empirically that parameter-space MAP is often able to perform on par with L-MAP in accuracy, although L-MAP tends to improve calibration.

The aim of this paper is to improve our understanding of what it means to learn most likely functions instead of parameters. Our analysis, which includes theoretical insights as well as experiments with both carefully designed tractable models and neural networks, paints a complex picture and unearths distinct benefits and drawbacks of learning most likely functions instead of parameters. We conclude with a discussion of practical considerations around function-space MAP estimation, its relationship to flat minima and generalization, and the practical relevance of parameterization invariance.

Our code is available at https://github.com/activatedgeek/function-space-map.

## 2 Preliminaries

We consider supervised learning problems with a training dataset $\mathcal{D} = (x_\mathcal{D}, y_\mathcal{D}) = \{x_\mathcal{D}^{(i)}, y_\mathcal{D}^{(i)}\}_{i=1}^N$ of inputs $x \in \mathcal{X}$ and targets $y \in \mathcal{Y}$ with input space $\mathcal{X} \subseteq \mathbb{R}^D$ and output space $\mathcal{Y} \subseteq \mathbb{R}^K$. For example, the inputs $x$ could correspond to times, spatial locations, tabular data, images, and the targets $y$ to regression values, class labels, etc.

To create a model of the data $\mathcal{D}$, one typically starts by specifying a function $f_\theta : \mathcal{X} \to \mathcal{Y}$ parametrized by a vector $\theta \in \mathbb{R}^P$ which maps inputs to outputs. $f_\theta$ could be a neural network, a polynomial model, a Fourier series, and so on. Learning typically amounts to estimating model parameters $\theta$. To this end, we can relate the function $f_\theta$ to the targets through an *observation model*, $p(y|x, f_\theta)$. For example, in regression, we could assume the outputs $y = f_\theta(x) + \epsilon$, where $\epsilon \sim \mathcal{N}(0, \sigma^2)$ is additive Gaussian noise with variance $\sigma^2$. Equivalently, $p(y|x, \theta) = \mathcal{N}(y|f_\theta(x), \sigma^2)$. Alternatively, in classification, we could specify $p(y|x, \theta) = \mathrm{Categorical}(\mathrm{softmax}(f_\theta(x)))$. We then use this observation model to form a *likelihood* over the whole dataset $p(y_\mathcal{D}|x_\mathcal{D}, \theta)$. In each of these example observation models, the likelihood factorizes, as the data points are conditionally independent given model parameters $\theta$, $p(y_\mathcal{D}|x_\mathcal{D}, \theta) = \prod_{i=1}^N p(y_\mathcal{D}^{(i)}|x_\mathcal{D}^{(i)}, \theta)$. We can further express a belief over values of parameters through a prior $p(\theta)$, such as $p(\theta) = \mathcal{N}(\theta|\mu, \Sigma)$. Finally, using Bayes rule, the log of the posterior up to a constant $c$ independent of $\theta$ is,

$$\log \overbrace{p(\theta|y_\mathcal{D}, x_\mathcal{D})}^{\text{parameter posterior}} = \log \overbrace{p(y_\mathcal{D}|x_\mathcal{D}, \theta)}^{\text{likelihood}} + \log \overbrace{p(\theta)}^{\text{prior}} + c. \tag{1}$$

Notably, standard loss functions are negative log posteriors, such that maximizing this parameter posterior, which we refer to as *parameter-space* maximum a posteriori (PS-MAP) estimation, corresponds to minimizing standard loss functions [19]. For example, if the observation model is regression with Gaussian noise or Laplace noise, the negative log likelihood is proportional to mean-square or mean-absolute error functions, respectively. If the observation model is a categorical distribution with a softmax link function, then the log likelihood is negative cross-entropy. If we use a zero-mean Gaussian prior, we recover standard $\ell_2$ regularization, also known as weight-decay. If we use a Laplace prior, we recover $\ell_1$ regularization, also known as LASSO [26].

Once we maximize the posterior to find

$$\hat{\theta}^{\text{PS-MAP}} = \arg\max_\theta p(\theta|y_\mathcal{D}, x_\mathcal{D}), \tag{2}$$

we can condition on these parameters to form our function $f_{\hat{\theta}^{\text{PS-MAP}}}$ to make predictions. However, as we saw in Figure 1, $f_{\hat{\theta}^{\text{PS-MAP}}}$ is not in general the function that maximizes the posterior over functions, $p(f_\theta|y_\mathcal{D}, x_\mathcal{D})$. In other words, if

$$\hat{\theta}^{\text{FS-MAP}} = \arg\max_\theta p(f_\theta|y_\mathcal{D}, x_\mathcal{D}) \tag{3}$$

then generally $f_{\hat{\theta}^{\text{PS-MAP}}} \neq f_{\hat{\theta}^{\text{FS-MAP}}}$. Naively, one can write the log posterior over $f_\theta$ up to the same constant $c$ as above as

$$\log \overbrace{p(f_\theta|y_\mathcal{D}, x_\mathcal{D})}^{\text{function posterior}} = \log \overbrace{p(y_\mathcal{D}|x_\mathcal{D}, f_\theta)}^{\text{likelihood}} + \log \overbrace{p(f_\theta)}^{\text{function prior}} + c, \tag{4}$$

where $p(y_\mathcal{D}|x_\mathcal{D}, f_\theta) = p(y_\mathcal{D}|x_\mathcal{D}, \theta)$, but just written in terms of the function $f_\theta$. The prior $p(f_\theta)$ however is a different function from $p(\theta)$, because we incur an additional Jacobian factor in this change of variables, making the posteriors also different.

We must take care in interpreting the quantity $p(f_\theta)$ since probability densities in infinite-dimensional vector spaces are generally ill-defined. While prior work [14, 23, 24, 25, 30] avoids this problem by considering a prior only over functions evaluated at a finite number of evaluation points, we provide a more general objective that enables the use of infinitely many evaluation points to construct a more informative prior and makes the relevant design choices of function-space MAP estimation more interpretable.

## 3  Understanding Function-Space Maximum A Posteriori Estimation

Function-space MAP estimation seeks to answer a fundamentally different question than parameter-space MAP estimation, namely, what is the most likely function under the posterior distribution over functions implied by the posterior distribution over parameters, rather than the most likely parameters under the parameter posterior.

To better understand the benefits and shortfalls of function-space MAP estimation, we derive a function-space MAP objective that generalizes the objective considered by prior work and analyze its properties both theoretically and empirically.

## 3.1 The Finite Evaluation Point Objective

Starting from Equation (4), Wolpert [30] proposed to instead find the MAP estimate for $f_\theta(\hat{x})$, the function evaluated at a finite set of points $\hat{x} = \{x_1, ..., x_M\}$, where $M < \infty$ can be chosen to be arbitrarily large so as to capture the behavior of the function to arbitrary resolution. This choice then yields the FS-MAP optimization objective

$$\mathcal{L}_{\text{finite}}(\theta; \hat{x}) = \sum_{i=1}^{N} \log p(y_\mathcal{D}^{(i)} | x_\mathcal{D}^{(i)}, f_\theta(\hat{x})) + \log p(f_\theta(\hat{x})), \tag{5}$$

where $p(f_\theta(\hat{x}))$ is a well-defined but not in general analytically tractable probability density function. (See Appendix B.1 for further discussion.) Let $P$ be the number of parameters $\theta$, and $K$ the number of function outputs. Assuming the set of evaluation points is sufficiently large so that $MK \geq P$, using a generalization of the change of variable formula that only assumes injectivity rather than bijectivity, Wolpert [30] showed that the prior density over $f(\hat{x})$ is given by

$$p(f_\theta(\hat{x})) = p(\theta) \det^{-1/2}(\mathcal{J}(\theta; \hat{x})), \tag{6}$$

where $\mathcal{J}(\theta; \hat{x})$ is a $P$-by-$P$ matrix defined by

$$\mathcal{J}_{\text{finite}}(\theta; \hat{x}) \doteq J_\theta(\hat{x})^\top J_\theta(\hat{x}) \tag{7}$$

and $J_\theta(\hat{x}) \doteq \partial f_\theta(\hat{x})/\partial \theta$ is the $MK$-by-$P$ Jacobian of $f_\theta(\hat{x})$, viewed as an $MK$-dimensional vector, with respect to the parameters $\theta$. Substituting Equation (6) into Equation (5) the function-space MAP objective as a function of the parameters $\theta$ can be expressed as

$$\mathcal{L}_{\text{finite}}(\theta; \hat{x}) = \sum_{i=1}^{N} \log p(y_\mathcal{D}^{(i)} | x_\mathcal{D}^{(i)}, f_\theta) + \log p(\theta) - \frac{1}{2} \log \det(\mathcal{J}(\theta; \hat{x})), \tag{8}$$

where we are allowed to condition directly on $f_\theta$ instead of $f_\theta(\hat{x})$ because by assumption $\hat{x}$ is large enough to uniquely determine the function. In addition to replacing the function, our true quantity of interest, with its evaluations at a finite set of points $\hat{x}$, this objective makes it unclear how we should select $\hat{x}$ and how that choice can be interpreted or justified in a principled way.

## 3.2 Deriving a More General Objective and its Interpretation

We now derive a more general class of function-space MAP objectives that allow us to use effectively infinitely many evaluation points and meaningfully interpret the choice of those points.

In general, when performing a change of variables from $v \in \mathbb{R}^n$ to $u \in \mathbb{R}^m$ via an injective map $u = \varphi(v)$, their probability densities relate as $p(u)\mathrm{d}\mu(u) = p(v)\mathrm{d}\nu(v)$, where $\mu$ and $\nu$ define respective volume measures. Suppose we let $\mathrm{d}\mu(u) = \sqrt{\det(g(u))}\mathrm{d}^m u$, the volume induced by an $M \times M$ metric tensor $g$, and $\mathrm{d}\nu(v) = \mathrm{d}^n v$, the Lebesgue measure, then we can write $\mathrm{d}\mu(u) = \sqrt{\det(\tilde{g}(v))}\mathrm{d}\nu(v)$, where the $N \times N$ metric $\tilde{g}(v) = J(v)^\top g(u)J(v)$ is known as the pullback of $g$ via $\varphi$ [6] and $J$ is the Jacobian of $\varphi$ [6]. As a result, we have $p(u) = p(v)\det^{-1/2}(J(v)^\top g(u)J(v))$. Applying this argument and identifying $u$ with $f_\theta(\hat{x})$ and $v$ with $\theta$, Wolpert [30] thereby establishes $p(f_\theta(\hat{x})) = p(\theta) \det^{-1/2}(J_\theta(\hat{x})^\top J_\theta(\hat{x}))$.

However, an important implicit assumption in the last step is that the metric in function space is Euclidean. That is, $g = I$ and the squared distance between $f_\theta(\hat{x})$ and $f_\theta(\hat{x}) + \mathrm{d}f_\theta(\hat{x})$ is $\mathrm{d}s^2 = \sum_{i=1}^{M} \mathrm{d}f_\theta(x_i)^2$, rather than the general case $\mathrm{d}s^2 = \sum_{i=1}^{M} \sum_{j=1}^{M} g_{ij}\mathrm{d}f_\theta(x_i)\mathrm{d}f_\theta(x_j)$. To account for a generic metric $g$, we therefore replace $J_\theta(\hat{x})^\top J_\theta(\hat{x})$ with $J_\theta(\hat{x})^\top g J_\theta(\hat{x})$. For simplicity, we assume the function output is univariate ($K = 1$) and only consider $g$ that is constant i.e., independent of $f_\theta(\hat{x})$ and diagonal, with $g_{ii} = g(x_i)$ for some function $g : \mathcal{X} \to \mathbb{R}^+$. For a discussion of the general case, see Appendix B.2. To better interpret the choice of $g$, we rewrite

$$J_\theta(\hat{x})^\top g J_\theta(\hat{x}) = \sum_{j=1}^{M} \tilde{p}_X(x_j)J_\theta(x_j)^\top J_\theta(x_j), \tag{9}$$

where we suggestively defined the alias $\tilde{p}_X(x) \doteq g(x)$ and $J_\theta(x_j)$ is the $K$-by-$P$-dimensional Jacobian evaluated at the point $x_j$. Generalizing from the finite and discrete set $\hat{x}$ to the possibly infinite entire domain $\mathcal{X} \subseteq \mathbb{R}^D$ and further dividing by an unimportant normalization constant $Z$, we obtain

$$\mathcal{J}(\theta; p_X) \doteq \frac{1}{Z} \int_\mathcal{X} \tilde{p}_X(x)J_\theta(x)^\top J_\theta(x) \, \mathrm{d}x = \mathbb{E}_{p_X}\left[J_\theta(X)^\top J_\theta(X)\right], \tag{10}$$

where $p_X = \tilde{p}_X/Z$ is a normalized probability density function, with normalization $Z$—which is independent of $\theta$—only appearing as an additive constant in $\log \det \mathcal{J}(\theta; p_X)$. We include a further discussion about this limit in Appendix B.4.

Under this limit, we can identify $\log p(\theta) - \frac{1}{2} \log \det \mathcal{J}(\theta; p_X)$ with $\log p(f_\theta)$ up to a constant, and thereby express $\log p(f_\theta | \mathcal{D})$, the function-space MAP objective in Equation (4) as

$$\mathcal{L}(\theta; p_X) = \sum_{i=1}^{N} \log p(y_{\mathcal{D}}^{(i)} | x_{\mathcal{D}}^{(i)}, f_\theta) + \log p(\theta) - \frac{1}{2} \log \det \mathcal{J}(\theta; p_X), \quad (11)$$

where the choice of $p_X$ corresponds to a choice of the metric $g$ in function space. Equation (11) (and approximations thereof) will be the primary object of interest in the remainder of this paper.

**Evaluation Distribution as Function-Space Geometry.** From this discussion and Equations (10) and (11), it is now clear that the role of the metric tensor $g$ is to impose a distribution $p_X$ over the evaluation points. Or conversely, a given distribution over evaluation points implies a metric tensor, and as such, specifies the geometry of the function space. This correspondence is intuitive: points $x$ with higher values of $g(x)$ contribute more to defining the geometry in function space and therefore should be assigned higher weights under the evaluation distribution when maximizing function space posterior density. Suppose, for example, $f_\theta$ is an image classifier for which we only care about its outputs on set of natural images $\mathcal{I}$ when comparing it to another image classifier. The metric $g(\cdot)$ and therefore the evaluation distribution $p_X$ only needs support in $\mathcal{I}$, and the FS-MAP objective is defined only in terms of the Jacobians evaluated at natural images $x \in \mathcal{I}$.

**Finite Evaluation Point Objective as a Special Case.** Consequently, the finite evaluation point objective in Equation (8) can be arrived at by specifying the evaluation distribution $p_X(x)$ to be $\hat{p}_X(x) \doteq \frac{1}{M} \sum_{x' \in \hat{x}} \delta(x - x')$, where $\delta$ is the Dirac delta function and $\hat{x} = \{x_1, ..., x_M\}$ with $M < \infty$, as before. It is easy to see that $\mathcal{J}_{\text{finite}}(\theta; \hat{x}) \propto \mathcal{J}(\theta; \hat{p}_X)$. Therefore, the objective proposed by Wolpert [30] is a special case of the more general class of objectives.

### 3.3 Investigating the Properties of Function-Space MAP Estimation

To illustrate the properties of FS-MAP, we consider the class of models $f_\theta(x) = \sum_{i=1}^{P} \sigma(\theta_i) \varphi_i(x)$ with domain $\mathcal{X} = [-1, 1]$, where $\{\varphi_i\}_{i=1}^{P}$ is a fixed set of basis functions and $\sigma$ is a non-linear function to introduce a difference between function-space and parameter-space MAP. The advantage of working with this class of models is that $\mathcal{J}(\theta; p_X)$ has a simple closed form, such that

$$\mathcal{J}_{ij}(\theta; p_X) = \mathbb{E}_{p_X}\left[ \partial_{\theta_i} f_\theta(X) \partial_{\theta_j} f_\theta(X) \right] = \sigma'(\theta_i) \sigma'(\theta_j) \Phi_{ij}, \quad (12)$$

where $\Phi$ is a constant matrix with elements $\Phi_{ij} = \mathbb{E}_{p_X}[\varphi_i(X) \varphi_j(X)]$. Therefore, $\Phi$ can be precomputed once and reused throughout training. In this experiment, we use the set of Fourier features $\{\cos(k_i \cdot), \sin(k_i \cdot)\}_{i=1}^{100}$ where $k_i = i\pi$ and set $\sigma = \tanh$. We generate training data by sampling $x_{\text{train}} \sim \text{Uniform}(-1, 1)$, $\theta_i \sim \mathcal{N}(0, \alpha^2)$ with $\alpha = 10$, evaluating $f_\theta(x_{\text{train}})$, and adding Gaussian noise with standard deviation $\sigma^* = 0.1$ to each observation. We use 1,000 test points sampled from $\text{Uniform}(-1, 1)$. To train the model, we set the prior $p(\theta)$ and the likelihood to correspond to the data-generating process. For FS-MAP, we specify $p_X = \text{Uniform}(-1, 1)$, the ground-truth distribution of test inputs, which conveniently results in $\Phi = I/2$.

**FS-MAP Finds More Probable Functions.** Figure 2a shows the improvement in $\log p(f_\theta | \mathcal{D})$ when using FS-MAP over PS-MAP. As expected, FS-MAP consistently finds functions with much higher posterior probability $p(f_\theta | \mathcal{D})$.

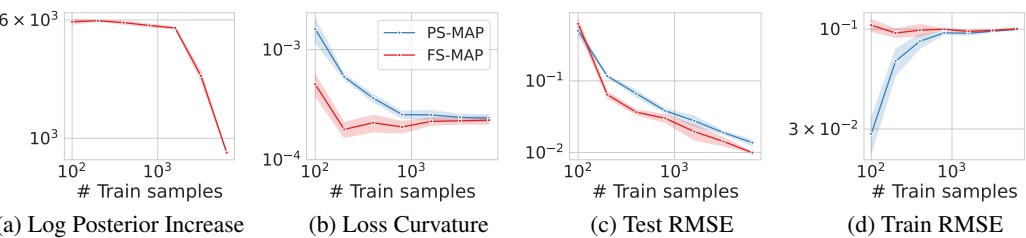

(a) Log Posterior Increase     (b) Loss Curvature     (c) Test RMSE     (d) Train RMSE

Figure 2: **FS-MAP exhibits desirable properties.** On a non-linear regression problem, FS-MAP empirically (a) learns more probable functions, (b) finds flatter minima, (c) improves generalization, and (d) is less prone to overfitting. The plot shows means and standard deviations computed from 3 random seeds.

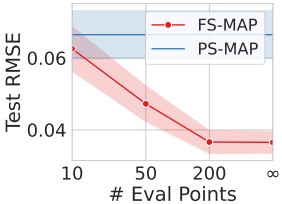
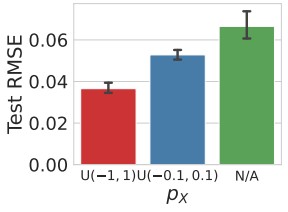

(a) Number of evaluation points          (b) Evaluation distribution

Figure 3: **Effect of important hyperparameters in FS-MAP.** FS-MAP improves with the number of evaluation points and requires carefully specifying the evaluation distribution and the probabilistic model.

**FS-MAP Prefers Flat Minima.** In Figure 2b, we compare the curvature of the minima found by FS-MAP and PS-MAP, as measured by the average eigenvalue of the Hessian of the mean squared error loss, showing that FS-MAP consistently finds flatter minima. As the objective of FS-MAP favors small singular values for the Jacobian $J_\theta$, when multiple functions can fit the data, the FS-MAP objective will generically prefer functions that are less sensitive to perturbations of the parameters, leading to flatter minima. To make this connection more precise, consider the Hessian $\nabla^2 \mathcal{L}(\theta)$. If the model fits the training data well, we can apply the Gauss-Newton approximation: $\nabla^2_\theta \mathcal{L}(\theta) \approx \frac{1}{|\mathcal{D}|} J_\theta(x_\mathcal{D})^\top J_\theta(x_\mathcal{D})$, which is identical to $\mathcal{J}(\theta; p_X)$ if $p_X$ is chosen to be the empirical distribution of the training inputs. More generally, a distribution $p_X$ with high density over likely inputs will assign high density to the training inputs, and hence minimizing $\mathcal{J}(\theta; p_X)$ will similarly reduces the magnitude of $J_\theta(x_\mathcal{D})$. Therefore, the FS-MAP objective explicitly encourages finding flatter minima, which have been found to correlate with generalization [10, 17, 18], robustness to data perturbations and noisy activations [3, 12] for neural networks.

**FS-MAP can Achieve Better Generalization.** Figure 2c shows that FS-MAP achieves lower test RMSE across a wide range of sample sizes. It's worth noting that this synthetic example satisfies two important criteria such that FS-MAP is likely to improve generalization over PS-MAP. First, the condition outlined in Section 4 for a well-behaved FS-MAP objective is met, namely that the set of partial derivatives $\{\partial_{\theta_i} f_\theta^j(\cdot)\}_{i=1}^P$ are linearly independent. Specifically, the partial derivatives are given by $\{\mathrm{sech}(\theta_i)\sin(k_i\cdot), \mathrm{sech}(\theta_i)\cos(k_i\cdot)\}_{i=1}^P$. Since $\mathrm{sech}$ is non-zero everywhere, the linear independence follows from the linear independence of the Fourier basis. As a result, the function space prior has no singularities and FS-MAP is thus able to learn from data. The second important criterion is the well-specification of the probabilistic model, which we have defined to precisely match the true data-generating process. Therefore, FS-MAP seeks the most likely function according to its *true* probability, without incorrect modeling assumptions.

**FS-MAP is Less Prone to Overfitting.** As shown in Figure 2d, FS-MAP tends to have a higher train RMSE than PS-MAP as a result of the additional log determinant regularization. While PS-MAP achieves near-zero train RMSE with a small number of samples by overfitting to the noise, FS-MAP's train RMSE is consistently around $\sigma^* = 0.1$, the true standard deviation of the observation noise.

**Performance Improves with Number of Evaluation Points.** We compare FS-MAP estimation with $p_X = \mathrm{Uniform}(-1, 1)$ and with a finite number of equidistant evaluation points in $[-1, 1]$, where the former corresponds to the latter with infinitely many points. In Figure 3a, we show that the test RMSE of the FS-MAP estimate (evaluated at 400 training samples) decreases monotonically until the number of evaluation points reaches 200. The more evaluation points, the more the finite-point FS-MAP objective approximates its infinite limit and the better it captures the behavior of the function. Indeed, 200 points is the minimum sampling rate required such that the Nyquist frequency reaches the maximum frequency in the Fourier basis, explaining the saturation of the performance.

**Choice of Evaluation Distribution is Important.** In Figure 3b, we compare test RMSE for 400 training samples for the default choice of $p_X = \mathrm{Uniform}(-1, 1)$, $p_X = \mathrm{Uniform}(-0.1, 0.1)$—a distribution that does not reflect inputs at test time—and PS-MAP ($p_X = \mathrm{N/A}$) for reference. The result shows that specifying the evaluation distribution to correctly reflect the distribution of inputs at test time is required for FS-MAP to achieve optimal performance in this case.

# 4 Limitations and Practical Considerations

We now discuss the limitations of function-space MAP estimation and propose partial remedies, including a scalable approximation that improves its applicability to modern deep learning.

## 4.1 FS-MAP Does not Necessarily Generalize Better than PS-MAP

*There is no guarantee that the most likely function is the one that generalizes the best,* especially since our prior can be arbitrary. For example, FS-MAP can under-fit the data if the log determinant introduces excessive regularization, which can happen if our probabilistic model is misspecified by overestimating the observation noise. Returning to the setup in Section 3.3, in Figure 4, we find that FS-MAP (with 400 training samples) is sensitive to the observation noise scale $\sigma$ in the likelihood. As $\sigma$ deviates from the true noise scale $\sigma^*$, the test RMSE of FS-MAP can change dramatically. At $\sigma/\sigma^* = 1/3$, the likelihood dominates in both the FS-MAP and PS-MAP objectives, resulting in similar test RMSEs. At $\sigma/\sigma^* = 10$, the likelihood is

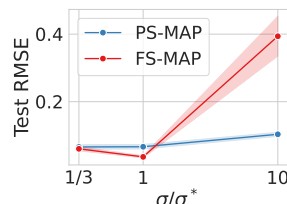

Figure 4: FS-MAP can underperform PS-MAP with a misspecified probabilistic model.

overpowered by the log determinant regularization in FS-MAP and causes the model to under-fit the data and achieve a high test error. By contrast, the performance of the PS-MAP estimate is relatively stable. Finally, even if our prior exactly describes the data-generating process, the function that best generalizes won't necessarily be the most likely under the posterior.

## 4.2 Pathological Solutions

In order for function-space MAP estimation to be useful in practice, the prior "density" $p(f_\theta) = p(\theta) \det^{-\frac{1}{2}}(\mathcal{J}(\theta; p_X))$ must not be infinite for any allowed values of $\theta$, since if it were, those values would constitute global optima of the objective function independent of the actual data. To ensure that there are no such pathological solutions, we require the matrix $\mathcal{J}(\theta; p_X)$ to be non-singular for any allowed $\theta$. We present two results that help determine if this requirement is met.

**Theorem 1.** *Assuming $p_X$ and $J_\theta = \partial_\theta f_\theta$ are continuous, $\mathcal{J}(\theta; p_X)$ is non-singular if and only if the partial derivatives $\{\partial_{\theta_i} f_\theta(\cdot)\}_{i=1}^P$ are linearly independent functions over the support of $p_X$.*
*Proof.* See Appendix B.5 ∎

**Theorem 2.** *If there exists a non-trivial symmetry $S \in \mathbb{R}^{P \times P}$ with $S \neq I$ such that $f_\theta = f_{S\theta}$ for all $\theta$, then $\mathcal{J}(\theta^*; p_X)$ is singular for all fixed points $\theta^*$ where $S\theta^* = \theta^*$.*
*Proof.* See Appendix B.6 ∎

Theorem 1 is analogous to $A^\top A$ having the same null space as $A$ for any matrix $A$. It suggests that to avoid pathological optima the effect of small changes to each parameter must be linearly independent. Theorem 2 builds on this observation to show that if the model exhibits symmetries in its parameterization, FS-MAP will necessarily have pathological optima. Since most neural networks at least possess permutation symmetries of the hidden units, we show that these pathological FS-MAP solutions are almost universally present, generalizing the specific cases observed by Wolpert [30].

**A Simple Remedy to Remove Singularities.** Instead of finding a point approximation of the function space posterior, we can perform variational inference under a variational family $\{q(\cdot|\theta)\}_\theta$, where $q(\cdot|\theta)$ is localized around $\theta$ with a small and constant *function-space* entropy $h$. Ignoring constants and $\mathcal{O}(h)$ terms, the variational lower bound is given by

$$\mathcal{L}_{\text{VLB}}(\theta; p_X) = \sum_{i=1}^N \log p(y_{\mathcal{D}}^{(i)}|x_{\mathcal{D}}^{(i)}, f_\theta) + \log p(\theta) - \frac{1}{2}\mathbb{E}_{q(\theta'|\theta)}\left[\log \det \mathcal{J}(\theta'; p_X)\right]. \quad (13)$$

Similar to how convolving an image with a localized Gaussian filter removes high-frequency components, the expectation $\mathbb{E}_{q(\theta'|\theta)}$ removes potential singularities in $\log \det \mathcal{J}(\theta'; p_X)$. This effect can be approximated by simply adding a small diagonal jitter $\epsilon$ to $\mathcal{J}(\theta; p_X)$ :

$$\hat{\mathcal{L}}(\theta; p_X) \doteq \sum_{i=1}^N \log p(y_{\mathcal{D}}^{(i)}|x_{\mathcal{D}}^{(i)}, f_\theta) + \log p(\theta) - \frac{1}{2}\log \det \left(\mathcal{J}(\theta; p_X) + \epsilon I\right). \quad (14)$$

Alternatively, we know $\mathbb{E}_{q(\theta'|\theta)}\left[\log \det \mathcal{J}(\theta'; p_X)\right]$ must be finite because the variational lower bound cannot exceed the log marginal likelihood. $\hat{\mathcal{L}}(\theta; p_X)$ can be used as a minimal modification of FS-MAP that eliminates pathological optima. We provide full derivation for these results in Appendix B.7.

## 4.3 Does the Function-Space MAP Better Approximate the Bayesian Model Average?

The Bayesian model average (BMA), $f_{\text{BMA}}$, of the function $f_\theta$ is given by the posterior mean, that is:

$$f_{\text{BMA}}(\cdot) \doteq \mathbb{E}_{p(\theta|\mathcal{D})}[f_\theta(\cdot)] = \mathbb{E}_{p(f_\theta|\mathcal{D})}[f_\theta(\cdot)].$$

This expectation is the same when computed in parameter or function space, and has theoretically desirable generalization properties [19, 29]. Both PS-MAP and FS-MAP provide point approximations of $f_{\text{BMA}}$. However, FS-MAP seeks the mode of the distribution $p(f_\theta|\mathcal{D})$, the mean of which we aim to compute. By contrast PS-MAP finds the mode of a distinct distribution, $p(\theta|\mathcal{D})$, which can markedly diverge from $p(f_\theta|\mathcal{D})$, depending on the parameterization. Consequently, it is reasonable to anticipate that the FS-MAP objective generally encourages finding a superior estimate of the BMA.

Consider the large data regime where the posterior in both parameter and function space follows a Gaussian distribution, in line with the Bernstein-von Mises theorem [27]. In Gaussian distributions, the mode and mean coincide, and therefore $f_{\hat{\theta}\text{FS-MAP}} = f_{\text{BMA}}$. However, even in this setting, where $\hat{\theta}^{\text{PS-MAP}} = \mathbb{E}_{p(\theta|\mathcal{D})}[\theta]$, generally $f_{\hat{\theta}\text{PS-MAP}} \neq f_{\text{BMA}}$, because the expectation does not distribute across a function $f$ that is non-linear in its parameters $\theta$: $f_{\mathbb{E}_{p(\theta|\mathcal{D})}[\theta]}(\cdot) \neq \mathbb{E}_{p(\theta|\mathcal{D})}[f_\theta(\cdot)]$.

However, we find that whether FS-MAP better approximates the BMA depends strongly on the problem setting. Returning to the setup in Figure 1, where we have a Gaussian mixture regression model, we compare the BMA function with functions learned by FS-MAP and PS-MAP under the prior $p(\theta) = \mathcal{N}(0, \alpha^2)$ with several settings of $\alpha$. We observe in Figure 5 that FS-MAP only approximates the BMA function better than PS-MAP at larger $\alpha$ values. To understand this behavior, recall the height $h_R$ for the right Gaussian bump is given by $\exp(\theta_R)$, which has a $\text{lognormal}(0, \alpha^2)$ prior. As we increase $\alpha$, more prior mass is assigned to $h_R$ with near-zero value and therefore to functions with small values within $x \in [0, 1]$. While the lognormal distribution also has a heavy tail at large $h_R$, the likelihood constrains the posterior $p(h_R|\mathcal{D})$ to only place high mass for small $h_R$. These two effects combine to make the posterior increasingly concentrated in function space around functions described by $h_L \approx 1$ and $h_R \approx 0$, implying that the mode in function space should better approximate its mean. In contrast, as we decrease $\alpha$, both the prior and posterior become more concentrated in parameter space since the parameter prior $p(\theta) = \mathcal{N}(0, \alpha^2)$ approaches the delta function at zero, suggesting that PS-MAP should be a good approximation to the BMA function. By varying $\alpha$, we can interpolate between how well PS-MAP and FS-MAP approximate the BMA.

## 4.4 Scalable Approximation for Large Neural Networks

In practice, the expectation $\mathcal{J}(\theta; p_X) = \mathbb{E}_{p_X}[J_\theta(X)^\top J_\theta(X)]$ is almost never analytically tractable due to the integral over $X$. We show in Appendix B.9 that a simple Monte Carlo estimate for $\mathcal{J}(\theta; p_X)$ with $S$ samples of $X$ can yield decent accuracy. For large neural networks, this estimator is still prohibitively expensive: each sample will require $K$ backward passes, taking a total of $\mathcal{O}(SKP)$ time. In addition, computing the resulting $SK$-by-$P$ determinant takes time $\mathcal{O}(SKP^2)$ (assuming $P \geq SK$). However, we can remedy the challenge of scalability by further leaning into the variational objective described in Equation (13) and consider the regime where $\epsilon \gg \lambda_i$ for all eigenvalues $\lambda_i$ of $\mathcal{J}(\theta; p_X)$. To first order in $\max_i \lambda_i/\epsilon$, we have $\log \det(\mathcal{J}(\theta; p_X)) + \epsilon I) = \frac{1}{2}\Delta_\psi d(\theta, \theta + \psi))\big|_{\psi=0}$ where $d(\theta, \theta') \doteq \mathbb{E}_{p_X}[\|f_\theta(X) - f_{\theta'}(X)\|^2]$ and $\Delta = \sum_{i=1}^P \partial_{\theta_i}^2$ denotes the Laplacian operator. Exploiting the identity $\frac{1}{2}\Delta_\psi d(\theta, \theta + \psi))\big|_{\psi=0} = \frac{1}{\beta^2}\mathbb{E}_{\psi \sim \mathcal{N}(0, \beta^2 I)}[d(\theta, \theta + \psi)] + \mathcal{O}(\beta^2)$ and choosing $\beta$ small enough, we obtain an accurate Monte Carlo estimator for the Laplacian

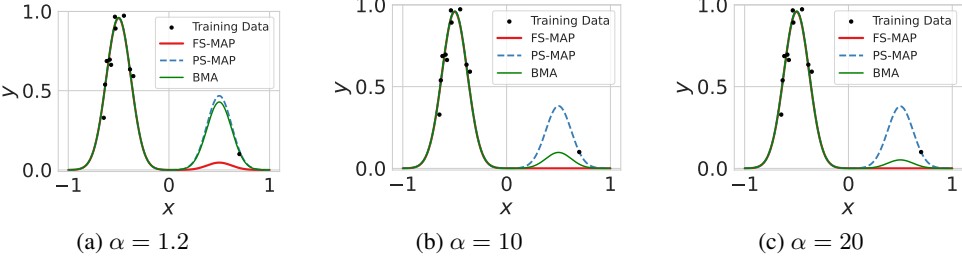

(a) $\alpha = 1.2$          (b) $\alpha = 10$          (c) $\alpha = 20$

Figure 5: FS-MAP does not necessarily approximate the BMA better than PS-MAP. A Gaussian mixture regression model, where the right Gaussian has weight $\exp(\theta_R)$ with prior $p(\theta_R) = \mathcal{N}(0, \alpha^2)$. As we increase $\alpha$, we interpolate between PS-MAP and FS-MAP approximating the BMA.

using only forward passes. The resulting objective which we refer to as Laplacian Regularized MAP (L-MAP) is given by

$$\mathcal{L}_{\text{L-MAP}}(\theta; p_X) \doteq \sum_{i=1}^{N} \log p(y_{\mathcal{D}}^{(i)} | x_{\mathcal{D}}^{(i)}, f_\theta) + \log p(\theta) - \frac{1}{2\epsilon\beta^2} \mathbb{E}_{\psi \sim \mathcal{N}(0,\beta^2 I)}[d(\theta, \theta + \psi)]. \quad (15)$$

We use a single sample of $\psi$ and $S$ samples of evaluation points for estimating $d(\theta, \theta + \psi)$ at each step, reducing the overhead from $\mathcal{O}(SKP^2)$ to only $\mathcal{O}(SP)$. A more detailed exposition is available in Appendix B.10.

### 4.5 Experimental Evaluation of L-MAP

We now evaluate L-MAP applied to neural networks on various commonly used datasets. We provide full experimental details and extended results in Appendix C. Unlike the synthetic task in Section 3.3, in applying neural networks to these more complex datasets we often cannot specify a prior that accurately models the true data-generating process, beyond choosing a plausible architecture whose high-level inductive biases align with the task (e.g. CNNs for images) and a simple prior $p(\theta)$ favoring smooth functions (e.g. an isotropic Gaussian with small variances). Therefore, we have less reason to expect L-MAP should outperform PS-MAP in these settings.

**UCI Regression.** In Table 1, we report normalized test RMSE on UCI datasets [1], using an MLP with 3 hidden layers and 256 units. L-MAP achieves lower error than PS-MAP on 7 out 8 datasets. Since the inputs are normalized and low-dimensional, we use $p_X = \mathcal{N}(0, I)$ for L-MAP.

Table 1: Normalized test RMSE ($\downarrow$) on UCI datasets. We report mean and standard errors over six trials.

| METHOD | BOSTON | CONCRETE | ENERGY | NAVAL | POWER | PROTEIN | WINERED | WINEWHITE |
|---|---|---|---|---|---|---|---|---|
| PS-MAP | **.329**$_{\pm.033}$ | .272$_{\pm.016}$ | .042$_{\pm.003}$ | .032$_{\pm.005}$ | .219$_{\pm.006}$ | .584$_{\pm.005}$ | .851$_{\pm.029}$ | .758$_{\pm.013}$ |
| L-MAP | .352$_{\pm.040}$ | **.261**$_{\pm.013}$ | **.041**$_{\pm.002}$ | **.018**$_{\pm.002}$ | **.218**$_{\pm.005}$ | **.580**$_{\pm.005}$ | **.792**$_{\pm.031}$ | **.714**$_{\pm.017}$ |

**Image Classification.** In Table 2, we compare L-MAP and PS-MAP on image classification using a ResNet-18 [9]. L-MAP achieves comparable or slightly better accuracies and is often better calibrated. We further test the effectiveness of L-MAP with transfer learning with a larger ResNet-50 trained on ImageNet. In Table 3, we show L-MAP also achieves small improvements in accuracy and calibration in transfer learning.

**Loss Landscape.** In Figure 6 (Left), we show that L-MAP indeed finds flatter minima. Further, we plot the Laplacian estimate in Figure 6 (Right) as the training progresses. We see that the Laplacian is much lower for L-MAP, showing its effectiveness at constraining the eigenvalues of $\mathcal{J}(\theta; p_X)$.

**Distribution of Evaluation Points.** In Table 2, we study the impact of the choice of distribution of evaluation points. Alongside our main choice of the evaluation set (KMNIST for FashionMNIST and CIFAR-100 for CIFAR-10), we use two additional distributions - the training set itself and a white noise $\mathcal{N}(0, I)$ distribution of the same dimensions as the training inputs. For both tasks, we find that using an external evaluation set beyond the empirical training distribution is often beneficial.

**Number of Evaluation Point Samples.** In Figure 7(a), we compare different Monte Carlo sample sizes $S$ for estimating the Laplacian. Overall, L-MAP is not sensitive to this choice in terms of accuracy. However, calibration error [20] sometimes monotonically decreases with $S$.

Table 2: We report the accuracy (ACC.), negative log-likelihood (NLL), expected calibration error [20] (ECE), and area under selective prediction accuracy curve [7] (SEL. PRED.) for FashionMNIST [31] ($\mathcal{D}'$ = KMNIST [4]), and CIFAR-10 [15] ($\mathcal{D}'$ = CIFAR-100). A ResNet-18 [9] is used. Std. errors are reported over five trials.

| METHOD | FASHIONMNIST | | | | CIFAR-10 | | | |
|---|---|---|---|---|---|---|---|---|
| | ACC.$\uparrow$ | SEL. PRED.$\uparrow$ | NLL$\downarrow$ | ECE$\downarrow$ | ACC.$\uparrow$ | SEL. PRED.$\uparrow$ | NLL$\downarrow$ | ECE$\downarrow$ |
| PS-MAP | 93.9%$_{\pm.1}$ | 98.6%$_{\pm.1}$ | .26$_{\pm.01}$ | 4.0%$_{\pm.1}$ | 95.4%$_{\pm.1}$ | 99.4%$_{\pm.0}$ | .18$_{\pm.00}$ | 2.5%$_{\pm.1}$ |
| L-MAP $p_X = \mathcal{N}(0, I)$ | 94.0%$_{\pm.0}$ | 99.2%$_{\pm.0}$ | .25$_{\pm.01}$ | 4.0%$_{\pm.2}$ | 95.3%$_{\pm.1}$ | 99.4%$_{\pm.0}$ | .20$_{\pm.00}$ | 3.0%$_{\pm.1}$ |
| $p_X = $TRAIN | 93.8%$_{\pm.1}$ | 99.2%$_{\pm.1}$ | .27$_{\pm.01}$ | 4.3%$_{\pm.2}$ | 95.6%$_{\pm.1}$ | 99.5%$_{\pm.0}$ | .18$_{\pm.01}$ | 2.6%$_{\pm.0}$ |
| $p_X = \mathcal{D}'$ | 94.1%$_{\pm.1}$ | 99.2%$_{\pm.1}$ | .26$_{\pm.01}$ | 4.1%$_{\pm.1}$ | 95.5%$_{\pm.1}$ | 99.5%$_{\pm.0}$ | .16$_{\pm.01}$ | 1.4%$_{\pm.1}$ |

Table 3: Transfer learning from ImageNet with ResNet-50 on CIFAR-10 can lead to slightly better accuracy and improved calibration.

| METHOD | ACC. ↑ | SEL. PRED. ↑ | NLL ↓ | ECE ↓ |
|--------|--------|--------------|-------|-------|
| PS-MAP | $96.3\%_{\pm 0.1}$ | $99.5\%_{\pm 0.1}$ | $0.18_{\pm 0.01}$ | $2.6\%_{\pm 0.2}$ |
| L-MAP | $96.4\%_{\pm 0.1}$ | $99.5\%_{\pm 0.1}$ | $0.15_{\pm 0.01}$ | $2.1\%_{\pm 0.1}$ |

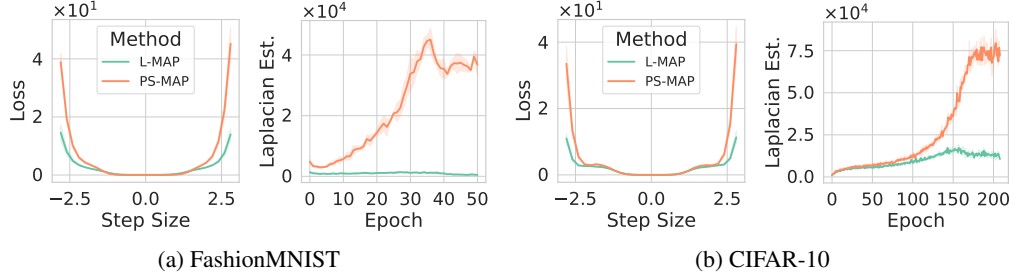

        (a) FashionMNIST                       (b) CIFAR-10

Figure 6: **Empirical evidence that L-MAP finds flatter minima. (Left)** For various step sizes in 20 randomly sampled directions starting at the minima, we compute the training loss averaged over all directions to summarize the local loss landscape. We use filter normalization for landscape visualization [16]. L-MAP visibly finds flatter minima. **(Right)** We plot the Laplacian estimate throughout training, showing L-MAP is indeed effective at constraining the eigenvalues of $\mathcal{J}(\theta; p_X)$.

**Robustness to Label Noise.** In Figure 7(b), we find L-MAP is slightly more robust to label noise than PS-MAP on CIFAR-10.

**Main Takeaways.** L-MAP shows qualitatively similar properties as FS-MAP such as favoring flat minima and often provides better calibration. However, it achieves comparable or only slightly better accuracy on more complex image classification tasks. In line with our expectations, these results suggest that accounting for the precise difference between FS-MAP and PS-MAP is less useful without a sufficiently well-motivated prior.

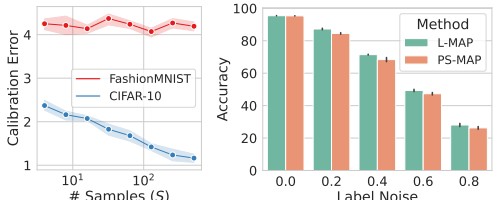

Figure 7: **(Left)** Calibration error can be reduced by increasing the number of Monte Carlo samples for the Laplacian estimator in Equation (15). **(Right)** L-MAP is slightly more robust to training with label noise. For each level of noise, we replace a fraction of labels with uniformly random labels.

## 5 Discussion

While we typically train our models through PS-MAP, we show that FS-MAP has many appealing properties in addition to re-parametrization invariance. We empirically verify these properties, including the potential for better robustness to noise and improved calibration. But we also reveal a more nuanced and unexpected set of pros and cons for each approach. For example, while it is natural to assume that FS-MAP more closely approximates a Bayesian model average, we clearly demonstrate how there can be a significant discrepancy. Moreover, while PS-MAP is not invariant to re-parametrization, which can be seen as a fundamental pathology, we show FS-MAP has its own failure modes such as pathological optima, as well as practical challenges around scalability. In general, our results suggest the benefits of FS-MAP will be greatest when the prior is sufficiently well-motivated.

Our findings engage with and contribute to active discussions across the literature. For example, several works have argued conceptually—and found empirically—that solutions in *flatter* regions of the loss landscape correspond to better generalization [8, 10, 11, 12, 25]. On the other hand, Dinh et al. [5] argue that the ways we measure flatness, for example, through Hessian eigenvalues, are not parameterization invariant, making it possible to construct striking failure modes. Similarly, PS-MAP estimation is not parameterization invariant. However, our analysis and comparison to FS-MAP estimation raise the question to what extent lack of parametrization invariance is actually a significant practical shortcoming—after all, we are not reparametrizing our models on the fly, and we have evolved our models and training techniques conditioned on standard parameterizations.

**Acknowledgements.** We thank Marc Finzi, Pavel Izmailov, and Micah Goldblum for helpful discussions. This work is supported by NSF CAREER IIS-2145492, NSF I-DISRE 193471, NSF IIS-1910266, BigHat Biosciences, Capital One, and an Amazon Research Award.

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

# Appendix

**Table of Contents**

# Appendix A    PS-MAP is not Invariant to Reparametrization

Parameter-space MAP estimation has conceptual and theoretical shortcomings stemming from the lack of reparameterization-invariance of MAP estimation. Suppose we reparameterize the model parameters $\theta$ with $\theta' = \mathcal{R}(\theta)$ where $\mathcal{R}$ is an invertible transformation. The prior density on the reparameterized parameters is obtained from the change-of-variable formula that states $p'(\theta') = p(\theta)|\det^{-1}(\mathrm{d}\theta'/\mathrm{d}\theta)|$. In the language of differential geometry, the prior density $p(\theta)$ is not a scalar but a scalar density, a quantity that is not invariant under coordinate transformations [28]. The probabilistic model is fundamentally unchanged, as we are merely viewing the parameters in a different coordinate system, but the MAP objective is not invariant to reparameterization. Specifically, the reparameterized parameter-space MAP objective becomes

$$\mathcal{L}'^{\mathrm{MAP}}(\theta') = \sum\nolimits_{i=1}^{N} \log p(y_{\mathcal{D}}^{(i)} \mid x_{\mathcal{D}}^{(i)}, \theta') + \log p'(\theta') = \mathcal{L}^{\mathrm{MAP}}(\theta) - \underbrace{\log|\det^{-1}(\mathrm{d}\theta'/\mathrm{d}\theta)|}_{\text{new term}}. \quad \text{(A.1)}$$

Since $\mathcal{L}'^{\mathrm{MAP}}(\theta')$ and the original objective $\mathcal{L}^{\mathrm{MAP}}(\theta)$ differ by a term which is non-constant in the parameters if $T$ is non-linear, the maxima $\theta'^{\mathrm{MAP}} \doteq \arg\max_{\theta'} \mathcal{L}'^{\mathrm{MAP}}(\theta')$ and $\theta^{\mathrm{MAP}} \doteq \arg\max_{\theta} \mathcal{L}^{\mathrm{MAP}}(\theta)$ under the two objectives will be different. Importantly, by "different" we don't just mean $\theta'^{\mathrm{MAP}} \neq \theta^{\mathrm{MAP}}$ but $\theta'^{\mathrm{MAP}} \neq \mathcal{R}(\theta^{\mathrm{MAP}})$. That is, they are not simply each other viewed in a different coordinate system but actually represent different functions. More accurately, therefore, one should say MAP estimation is not *equivariant* (rather than invariant) under reparameterization. As a result, when using a parameter-space MAP estimate to make predictions at test time, the predictions can change dramatically simply depending on how the model is parameterized. In fact, one can reparameterize the model so that PS-MAP will return an arbitrary solution $\theta_0$ irrespective of the observed data, by choosing a reparameterization $\theta' = \mathcal{R}(\theta)$ such that $\log|\det^{-1}(\mathrm{d}\theta'/\mathrm{d}\theta)|_{\theta=\theta_0} = -\infty$. One such choice is $\theta' = (\theta - \theta_0)^3$, where the exponent is taken element-wise.

As a less extreme example, consider the reparameterization $\theta' = 1/\theta$. A Gaussian prior on the original parameters $p(\theta) \propto \exp(-\theta^2/2)$ translates to a prior $p'(\theta') \propto \exp(-1/2\theta'^2)/\theta'^2 = \theta^2 \exp(-\theta^2/2)$ on the inverted parameters. Note the transformed prior, when mapped onto the original parameters $\theta$, assigns a higher weight on non-zero values of $\theta$ due to the quadratic factor coming from the log determinant. We illustrate this effect with a simple experiment where a linear model with RBF features is trained with a Gaussian likelihood and a Gaussian prior $\mathcal{N}(0, I)$ using PS-MAP to fit noisy observations from a simple function. Figure 8 show the predictions and learned weights when PS-MAP is performed both in the original parameterization and in the inverted parameterization. While PS-MAP favors small weights in the original parameterization, it favors non-zero weights in the inverted one, learning a less smooth function composed of a larger number of RBF bases.

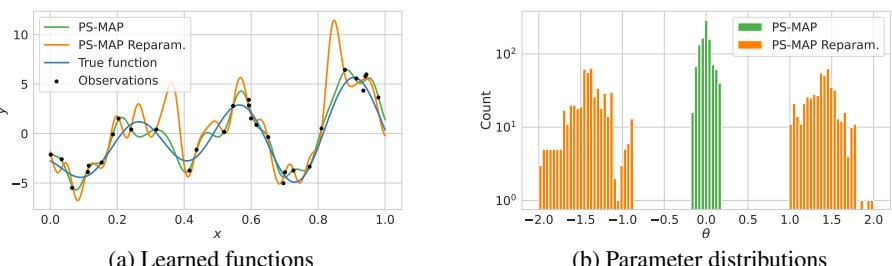

(a) Learned functions          (b) Parameter distributions

Figure 8: **PS-MAP is not invariant to reparameterization.** Simply inverting the parameterization of the weights removes the preference for small coefficients in a linear model, leading to a less smooth learned function, even though the underlying probabilistic model is unchanged.

# Appendix B   Mathematical Details & Derivations

## B.1   Prior Distributions over Functions

Since the function evaluations $f_\theta(\hat{x})$ are related to the parameters $\theta$ through the map $T : \theta \mapsto f_\theta(\hat{x})$, the prior density $p(f_\theta(\hat{x}))$ is simply the push forward of the density $p(\theta)$ through $T$, which can be expressed without loss of generality using the Dirac delta function as

$$p(f_\theta(\hat{x})) = \int_{\mathbb{R}^P} p(\theta')\delta(T(\theta') - f_\theta(\hat{x}))\, \mathrm{d}\theta'. \tag{B.2}$$

This generic expression as an integral over the parameter space is difficult to evaluate and thus not directly useful. By assuming that $T$ is injective (one-to-one) modulo some global symmetries in parameter space (e.g., permutation of the neurons), which holds for many neural network architectures [2], Wolpert [30] established the simplification in Equation (6). Note that since the map $T$ is not guaranteed to be surjective (onto), especially when there are more evaluation points than parameters ($|\hat{x}| > P$), $p(f_\theta(\hat{x}))$ does not generally have support everywhere and is to be interpreted as a surface density defined over the image of $T$ [30]. If the model parameterization has continuous symmetries, such as scaling symmetries in MLPs with ReLU activations [5], then the log determinant in Equation (6) should be replaced by the log pseudo-determinant to account for the fact that the dimension of the image of $T$ is less than $P$, the number of parameters.

## B.2   The General Case

Here we present a more general result for which the result in Section 3.2 is a special case, without assuming the function output is univariate ($K = 1$) or that the metric is diagonal. To simplify notations, we suppress all $\theta$-dependence and write $f_\theta^k(x_i)$, the $k$-th output of the function at $x_i$, as $f_i^k$ for some $\hat{x} = \{x_i\}_{i=1}^M$. We use Einstein notation to imply summation over repeated indices and leave everything in un-vectorized forms. In this more general case, the metric is given by $\mathrm{d}s^2 = g_{ikj\ell}\mathrm{d}f_i^k\mathrm{d}f_j^\ell$, where $i, j$ are input indices and $k, \ell$ are output indices. Denote the un-vectorized $M$-by-$K$-by-$P$ Jacobian $J_{ika} \doteq \partial_{\theta_a} f_i^k$. The $P$-by-$P$ matrix $\mathcal{J}$ appearing inside the determinant in Equation (6) is now given by

$$\mathcal{J}_{ab} = J_{ika}g_{ikj\ell}J_{j\ell b} = g_{ikj\ell}\partial_{\theta_a} f_i^k\partial_{\theta_b} f_j^\ell \tag{B.3}$$

In the limit of infinitely many evaluation points uniformly populating the domain $\mathcal{X}$, the sum over $i, j$ becomes a double integral[2]

$$\mathcal{J}_{ab} = \int_{\mathcal{X}\times\mathcal{X}} \partial_{\theta_a} f_\theta^k(x)\partial_{\theta_b} f_\theta^\ell(x')g_{k\ell}(x, x')\, \mathrm{d}x\, \mathrm{d}x', \tag{B.4}$$

where we have re-written the metric $g_{ikj\ell}$ as $g_{k\ell}(x_i, x_j)$ where $g : \mathcal{X} \times \mathcal{X} \to \mathbb{R}^{K\times K}$ is the metric represented now as a matrix-valued function. After dividing by a constant $Z$, we can similarly write $\mathcal{J}_{ab}$ as the expectation

$$\mathcal{J}_{ab} = \mathbb{E}_{p_{XX'CC'}}\left[\partial_{\theta_a} f_\theta^C(X)\partial_{\theta_b} f_\theta^{C'}(X')\mathrm{sgn}(g_{CC'}(X, X'))\right], \tag{B.5}$$

where $p_{XX'CC'}(x, x', c, c') = |g_{cc'}(x, x')|/Z$ for some normalization constant $Z$ that only shifts the log determinant by a $\theta$-independent constant. Note Equation (10) is recovered if the metric is diagonal, that is $g_{c,c'}(x, x') \neq 0$ only if $c = c'$ and $x = x'$. We can further remove the assumption that $g$ is constant (independent of $f$), but at the cost of letting the normalization constant $Z$ depend on $f$ (or equivalently $\theta$), making it invalid to ignore $Z$ during optimization. While more general than the result presented in Section 3.2, specifying a (possibly $\theta$-dependent) non-diagonal metric adds a significant amount of complexity without being particularly well-motivated. Therefore we did not further investigate this more general objective.

---

[2]To be more accurate, an integral of the form $\int \phi(x)\, \mathrm{d}x$ is the limit of $\sum_{i=1}^M \phi(x_i)\Delta x$ with the extra factor $\Delta x = x_{i+1} - x_i$ ($\forall i$), as $M \to \infty$. However, accounting for this factor does not affect the final result in our case because again $\Delta x$ is independent of $\theta$.

### B.3 Why Should a Constant Metric Affect FS-MAP?

One may object that a constant metric $g$ should have no effect in the MAP estimate for $f_\theta$. However, the subtlety arises from the fact that $p(f_\theta)$ does not have support everywhere but is, in fact, a surface density defined on the image of $T$ (as defined in Appendix B.1), a curved manifold in function space which gives rise to locally preferred directions such that even a global linear transformation will change the density in a non-homogeneous way. If $T$ were instead surjective, then in Section 3.2 $J_\theta(\hat{x})$ would be a square matrix and we could write $\log \det\left(J_\theta(\hat{x})^\top g J_\theta(\hat{x})\right) = \log \det\left(J_\theta(\hat{x})^\top J_\theta(\hat{x})\right) + \log \det(g)$ and conclude that a constant metric indeed has no effect on the MAP estimate. Similarly, if the image of $T$ is not curved, meaning $J_\theta(\hat{x})$ is constant in $\theta$, then $\log \det\left(J_\theta(\hat{x})^\top g J_\theta(\hat{x})\right)$ would be constant in $\theta$ regardless of $g$ and the metric would have no effect.

### B.4 Comments on the Infinite Limit

In Section 3.2, we generalized the finite evaluation point objective from Equation (8) by considering the limit as the number of evaluation points $\hat{x}$ approaches infinity and the points cover the domain $\mathbb{R}^P$ uniformly and densely. This technique, known as the continuum limit [21], is commonly used in physics to study continuous systems with infinite degrees of freedom, by first discretizing and then taking the limit as the discretization scale approaches zero, thereby recovering the behavior of the original system without directly dealing with intermediate (possibly ill-defined) infinite-dimensional quantities. Examples include lattice field theories where the continuous spacetime is discretized into a lattice, and numerical analysis where differential equations are discretized and solved on a mesh, where the solution often converges to the continuous solution as the mesh size goes to zero.

In a similar vein, we utilize this technique to sidestep direct engagement with the ill-defined quantity $p(f_\theta)$. It may be possible to assign a well-defined value to $p(f_\theta)$ through other techniques, though we expect the result obtained would be consistent with ours, given the continuum limit has historically yielded consistent results with potentially more refined approaches in other domains.

### B.5 Condition for a Non-Singular $\mathcal{J}(\theta; p_X)$

*Proof.* Recall $\mathcal{J}(\theta; p_X) = \mathbb{E}_{p_X}\left[J_\theta(X)^\top J_\theta(X)\right]$. Suppose $\mathcal{J}(\theta; p_X)$ is singular. Then there is a vector $v \neq 0$ for which $0 = v^\top \mathcal{J}(\theta; p_X) v = \mathbb{E}_{p_X}\left[\|J_\theta(X)v\|_2^2\right] = \int_\mathcal{X} \|J_\theta(X)v\|_2^2 p_X(x)\mathrm{d}x = 0$. Since the integrand is non-negative and continuous by assumption, it is zero everywhere. Therefore, we have $\sum_{i=1}^P v_i \partial_{\theta_i} f_\theta(x) = 0$ for all $x \in \mathrm{supp}(p_X)$ for a non-zero $v$, showing $\{\partial_{\theta_i} f_\theta(\cdot)\}_{i=1}^P$ are linearly dependent functions over the support of $p_X$.

Conversely, if $\{\partial_{\theta_i} f_\theta(\cdot)\}_{i=1}^P$ are linearly dependent functions over the support of $p_X$, then $\sum_{i=1}^P v_i \partial_{\theta_i} f_\theta(x) = J_\theta(x)v = 0$ for all $x \in \mathrm{supp}(p_X)$ for some $v \neq 0$. Hence $\mathcal{J}(\theta; p_X)v = \mathbb{E}_{p_X}\left[J_\theta(X)^\top J_\theta(X)v\right] = 0$, showing $\mathcal{J}(\theta; p_X)$ is singular. ∎

### B.6 Architectural Symmetries Imply Singular $\mathcal{J}(\theta; p_X)$

*Proof.* Differentiating with respect to $\theta$, we have $J_\theta = \partial_\theta f_\theta = \partial_\theta f_{S\theta} = J_{S\theta}S$ for all $\theta$. Suppose there exists $\theta^*$ that is invariant under $S$, $S\theta^* = \theta^*$, then $J_{\theta^*} = J_{\theta^*}S$ and $J_{\theta^*}(I - S) = 0$. Since, by assumption $S \neq I$, we have shown that $J_{\theta^*}$ has a non-trivial nullspace. Because the nullspace of $\mathcal{J}(\theta^*; p_X)$ contains the nullspace of $J_{\theta^*}$, we conclude $\mathcal{J}(\theta; p_X)$ is also singular. ∎

As an example, consider a single hidden layer MLP $f_\theta(x) = w_2^\top \sigma(w_1 x)$, $\theta = \{w_1 \in \mathbb{R}^2, w_2 \in \mathbb{R}^2\}$, where we neglected the bias for simplicity. The permutation $P_{12} \bigoplus P_{12}$ is a symmetry of $f_\theta$, where $\bigoplus$ is the direct sum and $P_{12}$ is the transposition $\begin{pmatrix} 0 & 1 \\ 1 & 0 \end{pmatrix}$. The nullspace of $J_{\theta^*}$ contains the image of

$$\begin{pmatrix} 1 & -1 \\ -1 & 1 \end{pmatrix} \bigoplus \begin{pmatrix} 1 & -1 \\ -1 & 1 \end{pmatrix} \tag{B.6}$$

with a basis $\{(1, -1, 0, 0), (0, 0, 1, -1)\}$, for any $\theta^*$ of the form $(a, a, b, b)$. It's easy to check that perturbing the parameters in directions $(1, -1, 0, 0)$ and $(0, 0, 1, -1)$ leaves the function output unchanged, due to symmetry in the parameter values and the network topology.

Continuous symmetries, among other reasons, can also lead to singular $\mathcal{J}(\theta; p_X)$ and are not covered by the above analysis, which is specific to discrete symmetries. One example is scaling symmetries in MLPs with ReLU activations [5]. However, these scaling symmetries cause $\mathcal{J}(\theta; p_X)$ to be singular for all $\theta$ in a manner that does not introduce any pathological optimum. This becomes clear when noting that the space of functions implemented by an MLP with $P$ parameters using ReLU activations lies in a space with dimension lower than $P$ due to high redundancies in the parameterization. The resolution is to simply replace all occurrences of $\log \det \mathcal{J}(\theta; p_X)$ with $\log |\mathcal{J}(\theta; p_X)|_+$, where $|\cdot|_+$ represents the pseudo-determinant. This step is automatic if we optimize Equation (14) instead of Equation (11), where adding a small jitter will automatically compute the log pseudo-determinant (up to an additive constant) when $\mathcal{J}(\theta; p_X)$ is singular. By contrast, symmetries such as permutation symmetries do create pathological optima because they only make the Jacobian singular at specific settings of $\theta$, thereby assigning to those points infinitely higher prior density compared to others.

## B.7 Addressing the Infinite Prior Density Pathology with a Variational Perspective

We now describe the remedy that leads to the objective in Equation (13). As shown in Section 4.2, almost all commonly-used neural networks suffer from the pathology such that a singular Jacobian leads infinite prior density at the singularities. Such singularities arise because FS-MAP optimize for a point approximation for the function space posterior, equivalent to variational inference with the family of delta functions $\{q(f_{\theta'}|\theta) = \delta(f_{\theta'} - f_\theta)\}_\theta$.

We can avoid the singularities if we instead choose a variational family where each member is a distribution over $f$ localized around $f_\theta$ for some $\theta$. Namely, we consider the family $\mathcal{Q} \doteq \{q(f_{\theta'}|\theta)\}_{\theta \in \Theta}$ parameterized by a mean-like parameter $\theta$ and a small but fixed entropy in function space $H(q(f_{\theta'}|\theta)) = \mathbb{E}_{q(f_{\theta'}|\theta)}[-\log q(f_{\theta'}|\theta)] = h$ for all $\theta \in \Theta$. For convenience, we overload the notation and use $q(\theta'|\theta)$ to denote the pullback of $q(f_{\theta'}|\theta)$ to parameter space. The resulting variational lower bound is

$$
\begin{aligned}
&\mathcal{L}_{\mathrm{VLB}}(\theta; p_X) \\
&= \mathbb{E}_{q(f_{\theta'}|\theta)}\left[\log \frac{q(f_{\theta'}|\theta)}{p(f_\theta|\mathcal{D})}\right] \\
&= \mathbb{E}_{q(f_{\theta'}|\theta)}[-\log p(f_{\theta'}|\mathcal{D})] - \overbrace{H(q(f_{\theta'}|\theta))}^{\text{const.}} \\
&= \mathbb{E}_{q(f_{\theta'}|\theta)}[-\log p(\mathcal{D}|f_{\theta'})] + \mathbb{E}_{q(f_{\theta'}|\theta)}[-\log p(f_{\theta'})] + \text{const.} && \text{(Bayes' rule)} \\
&= \mathbb{E}_{q(\theta'|\theta)}[-\log p(\mathcal{D}|\theta')] + \mathbb{E}_{q(\theta'|\theta)}[-\log p(\theta')] + \frac{1}{2}\mathbb{E}_{q(\theta'|\theta)}[\log \det(\mathcal{J}(\theta'; p_X))] && \text{(Equation (11))} \\
&= -\log p(\mathcal{D}|f_\theta) - \log p(\theta) + \frac{1}{2}\mathbb{E}_{q(\theta'|\theta)}[\log \det(\mathcal{J}(\theta'; p_X))] + \mathcal{O}(h) + \text{const.},
\end{aligned}
$$

where in the last line we used the assumption that $q(\theta'|\theta)$ is localized around $\theta$ to write the expectation of $\log p(\mathcal{D}|\theta)$ and $\log p(\theta)$ as their values at $\theta' = \theta$ plus $\mathcal{O}(h)$ corrections (which we will henceforth omit since by assumption $h$ is tiny), because they vary smoothly as a function of $\theta$. More care is required to deal with the remaining expectation, since when $\mathcal{J}(\theta; p_X)$ is singular at $\theta$, $\log \det(\mathcal{J}(\theta; p_X))$ is infinite, but its expectation, $\mathbb{E}_{q(\theta'|\theta)}[\log \det(\mathcal{J}(\theta'; p_X))]$, must be finite (assuming $p(\theta)$ and $p(\mathcal{D}|\theta)$ are finite), given that $\mathcal{L}_{\mathrm{VLB}}(\theta; p_X)$ lower bounds the log marginal likelihood $\log p(\mathcal{D}) = \log \int p(\mathcal{D}|\theta)p(\theta)\mathrm{d}\theta < \infty$. As we will show in Appendix B.8, the effect of taking the expectation of the log determinant is similar to imposing a lower limit $\epsilon$ on the eigenvalue of $\mathcal{J}(\theta; p_X)$, which can be approximated by adding a jitter $\epsilon$ when computing the log determinant. This effect is similar to applying a high-frequency cutoff to an image by convolving it with a localized Gaussian filter.

With this approximation, we arrive at a simple objective inspired by such a variational perspective, equivalent to adding a jitter inside the log determinant computation in the FS-MAP objective,

$$
\hat{\mathcal{L}}(\theta; p_X) = \sum_{i=1}^{N} \log p(y_\mathcal{D}^{(i)}|x_\mathcal{D}^{(i)}, \theta) + \log p(\theta) - \frac{1}{2}\log \det(\mathcal{J}(\theta; p_X) + \epsilon I). \tag{B.7}
$$

Note that while using a jitter is common in practice for numerical stability, it arises for an entirely different reason here.

## B.8 Approximating the Expectation

Rewriting $\log \det(\mathcal{J}(\theta'; p_X))$ in terms of the eigenvalues $\lambda_i(\theta')$, we have

$$\mathbb{E}_{q(\theta'|\theta)}[\log \det(\mathcal{J}(\theta'; p_X))] = \sum_{i=1}^{P} \mathbb{E}_{q(\theta'|\theta)}[\log \lambda_i(\theta')]. \tag{B.8}$$

Consider each term $\mathbb{E}_{q(\theta'|\theta)}[\log \lambda_i(\theta')]$ in isolation. Since the distribution $q(\theta'|\theta)$ is highly localized by assumption, there are only two regimes. If $\lambda_i(\theta) \neq 0$, then the variation of the log eigenvalue is small and the expectation is well approximated by $\log \lambda_i(\theta)$. Otherwise, if $\lambda_i(\theta) = 0$, then $\log \lambda_i(\theta')$ has a singularity at $\theta' = \theta$ and $\mathbb{E}_{q(\theta'|\theta)}[\log \lambda_i(\theta')] \approx \log \lambda_i(\theta) = -\infty$ is no longer a valid approximation since we've shown the expectation is finite. Therefore, in this case, the expectation must evaluate to a large but finite quantity, say $\log(\epsilon)$ for some small $\epsilon$, where the exact value of $\epsilon$ depends on the details of $q(\theta'|\theta)$ and $\lambda_i(\theta')$. Consequently, taking the expectation can be approximated by adding a jitter $\epsilon I$ to $\mathcal{J}(\theta'; p_X)$ when computing the log determinant to clip its eigenvalues from below at some threshold $\epsilon > 0$.

## B.9 Monte Carlo Estimator for the Log Determinant

**Monte Carlo Estimator.** Exactly computing $\mathcal{J}(\theta; p_X) = \mathbb{E}_{p_X}\left[J_\theta(X)^\top J_\theta(X)\right]$ and its log determinant is often intractable. Instead, we can use a simple Monte Carlo estimator to approximate the expectation inside the log determinant:

$$\log \det \left(\mathbb{E}_{p_X}\left[J_\theta(X)^\top J_\theta(X)\right]\right) \approx \log \det \left(\frac{1}{S} \sum_{j=1}^{S} J_\theta(x^{(j)})^\top J_\theta(x^{(j)}) + \epsilon I\right), \tag{B.9}$$

where we add a small amount of jitter to prevent the matrix inside the determinant from becoming singular when $SK < P$. Since $\frac{1}{S} \sum_{j=1}^{M} J_\theta(x^{(j)})^\top J_\theta(x^{(j)}) = \frac{1}{S} J_\theta(\hat{x}_S)^\top J_\theta(\hat{x}_S)$ where $\hat{x}_S = \{x^{(j)}\}_{j=1}^{S}$ and $J_\theta(\hat{x}_S)$ is an $SK$-by-$P$ matrix, one can compute the determinant of $\frac{1}{S} J_\theta(\hat{x}_S)^\top J_\theta(\hat{x}_S) + \epsilon I$ through the product of squared singular values of $J_\theta(\hat{x}_S)/\sqrt{S}$ in $\mathcal{O}(\min(SKP^2, S^2K^2P))$ time (with zero singular values replaced with $\sqrt{\epsilon}$), much faster than $\mathcal{O}(P^3)$ for computing the original $P$-by-$P$ determinant if $S$ is small enough, without ever storing the $P \times P$ matrix $\mathcal{J}(\theta; p_X)$.

**Accuracy of the Estimator.** Due to the nonlinearity in taking the log determinant, the estimator is not unbiased. To test its accuracy, we evaluate the estimator with $S = \{800, 400, 200\}$ for a neural network with two inputs, two outputs, four hidden layers, and 898 parameters, using $\tanh$ activations. The network is chosen small enough so that computing the exact log determinant is feasible. In Figure 9, we compare the exact log determinant for $p_X = \frac{1}{M} \sum_{i=1}^{M} \delta_{x_i}$ with $M = 1,600$, and its estimate with $S$ Monte Carlo samples. Here $\{x_i\}$ are linearly spaced in the region $[-5,5]^2$. We observe that using $S = 800$ can almost perfectly approximate the exact log determinant. With $S = 400$ and $S = 200$, the approximation underestimates the exact value because the number of zero singular values increases for $\frac{1}{S} J_\theta(\hat{x}_S)^\top J_\theta(\hat{x}_S)$, but it still maintains a strong correlation and monotonic relation with the exact value. We see that it is possible to approximate the log-determinant with sufficient accuracy with simple Monte Carlo using a fraction of the compute and memory for exact evaluation, though there can be a larger discrepancy as we continue to decrease $S$.

## B.10 Laplacian Regularized MAP Objective

For $\epsilon$ that is large enough compared to the eigenvalues of $\mathcal{J}(\theta; p_X)$), a first order approximation to $\log \det(\mathcal{J}(\theta; p_X) + \epsilon I)$ can be sufficiently accurate. Expanding to first order in $\rho \doteq \max_i \lambda_i/\epsilon$, we have

$$\log \det(\mathcal{J}(\theta; p_X) + \epsilon I) = \frac{1}{\epsilon} \sum_{i=1}^{P} \lambda_i + c_\epsilon + \mathcal{O}(\rho^2) = \frac{1}{\epsilon} \operatorname{Tr}(\mathcal{J}(\theta; p_X)) + c_\epsilon + \mathcal{O}(\rho^2), \tag{B.10}$$

where $c_\epsilon = P \log(\epsilon)$ is independent of $\theta$. Defining $d(\theta, \theta') \doteq \mathbb{E}_{p_X}[\|f_\theta(X) - f_{\theta'}(X)\|^2]$, we have

$$d(\theta, \theta + \psi) = \mathbb{E}_{p_X}[\|J_\theta(X)\psi\|^2] + \mathcal{O}(\psi^4) = \psi^\top \mathcal{J}(\theta; p_X))\psi + \mathcal{O}(\psi^4), \tag{B.11}$$

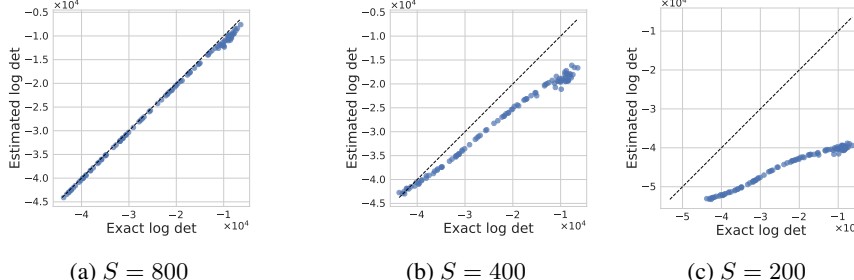

(a) $S = 800$        (b) $S = 400$        (c) $S = 200$

Figure 9: **Approximating the log determinant via an $S$-sample simple Monte Carlo estimator and jitter.** Each dot represents the exact and estimated log determinant evaluated at random parameters sampled by randomly initializing the network followed by scaling by a factor $s \sim \mathrm{loguniform}(0.1, 10)$ to include parameters of different magnitudes. The dashed line shows $x = y$.

which shows $\mathcal{J}(\theta; p_X)) = \frac{1}{2} \nabla_\psi^2 d(\theta, \theta + \psi)\big|_{\psi=0}$. Therefore,

$$\mathrm{Tr}(\mathcal{J}(\theta; p_X)) = \frac{1}{2} \mathrm{Tr}\big(\nabla_\psi^2 d(\theta, \theta + \psi)\big)\big|_{\psi=0} = \frac{1}{2} \Delta_\psi d(\theta, \theta + \psi))\big|_{\psi=0}, \tag{B.12}$$

where $\Delta$ is the Laplacian operator.

To estimate the Laplacian, consider the following expectation:

$$\mathbb{E}_{\psi \sim \mathcal{N}(0, \beta^2 I)}[d(\theta, \theta + \psi)] \tag{B.13}$$

$$= \mathbb{E}_{\psi \sim \mathcal{N}(0, \beta^2 I)}\big[\psi^\top \mathcal{J}(\theta; p_X))\psi + \mathcal{O}\big(\psi^4\big)\big] \tag{B.14}$$

$$= \mathbb{E}_{\psi \sim \mathcal{N}(0, \beta^2 I)}\big[\psi^\top \mathcal{J}(\theta; p_X))\psi\big] + \mathcal{O}\big(\beta^4\big) \tag{B.15}$$

$$= \beta^2 \, \mathrm{Tr}(\mathcal{J}(\theta; p_X)) + \mathcal{O}\big(\beta^4\big), \tag{B.16}$$

where we used

$$\mathbb{E}_\psi\big[\psi^\top \mathcal{J}(\theta; p_X))\psi\big] = \sum_{ij} \mathbb{E}_\psi[\psi_i \psi_j] \mathcal{J}_{ij}(\theta; p_X) = \beta^2 \sum_{ij} \delta_{ij} \mathcal{J}_{ij}(\theta; p_X) = \beta^2 \, \mathrm{Tr}(\mathcal{J}(\theta; p_X)). \tag{B.17}$$

Therefore, we have

$$\mathrm{Tr}(\mathcal{J}(\theta; p_X)) = \frac{1}{2} \Delta_\psi d(\theta, \theta + \psi))\big|_{\psi=0} = \frac{1}{\beta^2} \mathbb{E}_{\psi \sim \mathcal{N}(0, \beta^2 I)}[d(\theta, \theta + \psi)] + \mathcal{O}\big(\beta^2\big). \tag{B.18}$$

Combining Equation (B.10) and Equation (B.18), we have shown Equation (14) reduces to the more efficiently computable L-MAP objective for large enough $\epsilon$ and small enough $\beta$ :

$$\mathcal{L}_{\text{L-MAP}}(\theta; p_X) \doteq \sum_{i=1}^N \log p(y_\mathcal{D}^{(i)} \mid x_\mathcal{D}^{(i)}, f_\theta) + \log p(\theta) - \frac{1}{2\epsilon\beta^2} \mathbb{E}_{\psi \sim \mathcal{N}(0, \beta^2)}[d(\theta, \theta + \psi)]. \tag{B.19}$$

The entire objective is negated and divided by $N$ to yield the loss function

$$L_{\text{L-MAP}}(\theta; p_X) \doteq \underbrace{-\frac{1}{N} \sum_{i=1}^N \log p(y_\mathcal{D}^{(i)} \mid x_\mathcal{D}^{(i)}, f_\theta) - \frac{1}{N} \log p(\theta)}_{\text{Standard regularized loss}} + \lambda \underbrace{\left(\frac{1}{\beta^2} \mathbb{E}_{\psi \sim \mathcal{N}(0, \beta^2)}[d(\theta, \theta + \psi)]\right)}_{\text{Laplacian regularization } R(\theta; \beta)}, \tag{B.20}$$

where we have absorbed the $1/N$ factor into the hyperparameter $\lambda = \frac{1}{2\epsilon N}$. Therefore, using L-MAP amounts to simply adding a regularization $\lambda R(\theta; \beta^2)$ to standard regularized training (PS-MAP). $\beta$ is a tolerance parameter for approximating the Laplacian and can simply be fixed to a small value such as $10^{-3}$.

# Appendix C    Further Empirical Results and Experimental Details

## C.1    Training details for Synthetic Experiments

For both PS-MAP and FS-MAP, we train with the Adam [13] optimizer with a learning rate $0.1$ for $2,500$ steps to maximize the respective log posteriors. For FS-MAP, we precompute the $\Phi$ matrix and reuse it throughout training. When $p_X$ is not $\text{Uniform}(-1, 1)$, $\Phi$ does not have a simple form and we use $10,000$ Monte Carlo samples to evaluate it.

## C.2    Hyperparameters for UCI Regression

We use an MLP with 3 hidden layers, 256 units, and ReLU activations. We train it with the Adam optimizer for $10,000$ steps with a learning rate of $10^{-3}$. For each dataset, we tune hyperparameters based on validation RMSE, where the validation set is constructed by holding out $10\%$ of training data. We tune both the weight decay (corresponding to prior precision) and L-MAP's $\lambda$ over the choices $\{10^{-5}, 10^{-4}, 10^{-3}, 10^{-2}, 10^{-1}\}$. We then report the mean and standard error of test RMSE across 6 runs using the best hyperparameters selected this way. In each dataset, the inputs and outputs are standardized based on training statistics.

## C.3    Hyperparameters for Image Classification Experiments

Both the weight decay scale, corresponding to the variance of a Gaussian prior, and the L-MAP hyperparameter $\lambda$ in Equation (B.20) is tuned over the range $[10^{-1}, 10^{-10}]$ using randomized grid search. For PS-MAP, $\lambda$ is set to 0. The parameter variance in the Laplacian estimator is fixed to $\beta^2 = 10^{-6}$. In addition, we clip the gradients to unit norm. We use a learning rate of $0.1$ with SGD and a cosine decay schedule over 50 epochs for FashionMNIST and 200 epochs for CIFAR-10. The mini-batch size is fixed to 128.

## C.4    Verifying the L-MAP Approximation

To verify that the L-MAP objective is a good approximation to Equation B.7 when $\epsilon$ is large enough compared to the eigenvalues of $\mathcal{J}(\theta; p_X)$, we compare $\log \det(\mathcal{J}(\theta; p_X) + \epsilon I) - P \log(\epsilon)$ with the Laplacian estimate $\frac{1}{2\epsilon\beta^2} \mathbb{E}_{\psi \sim \mathcal{N}(0, \beta^2)}[d(\theta, \theta + \psi)]$ used by L-MAP in Appendix B.10 as its first order approximation in $\max_i \lambda_i(\theta)/\epsilon$. We reuse the same network architecture, evaluation distribution $p_X$, and sampling procedure for the parameters from Appendix B.9 to perform the comparison in Figure 10 and color each point by $\bar{\lambda}(\theta)/\epsilon$, the average eigenvalue of $\mathcal{J}(\theta; p_X)$ divided by $\epsilon$. The smaller this value, the better the approximation should be. We observe that indeed as $\epsilon$ increases and $\bar{\lambda}(\theta)/\epsilon$ approaches 0, the Laplacian estimate becomes a more accurate approximation of the log determinant. Interestingly, even when $\epsilon \ll \bar{\lambda}(\theta)$ and the approximation is not accurate, there still appears to be a monotonic relation between the estimate and the log determinant, suggesting that the Laplacian estimate will continue to produce a qualitatively similar regularization effect as the log determinant in that regime.

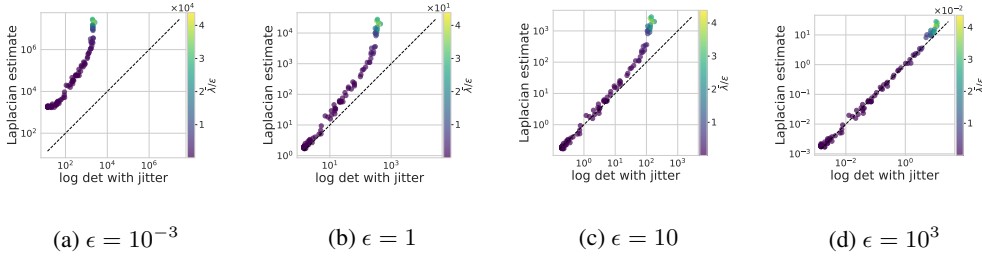

(a) $\epsilon = 10^{-3}$        (b) $\epsilon = 1$        (c) $\epsilon = 10$        (d) $\epsilon = 10^3$

Figure 10: **Log determinant v.s. the Laplacian estimate at different values of** $\epsilon$. The $x$-axis shows $\log \det(\mathcal{J}(\theta; p_X) + \epsilon I) - P \log(\epsilon)$ while the $y$-axis shows its first order approximation $\frac{1}{2\epsilon\beta^2} \mathbb{E}_{\psi \sim \mathcal{N}(0, \beta^2)}[d(\theta, \theta + \psi)]$, estimated with 10 samples of $\psi$.

## C.5    Effective Eigenvalue Regularization using the Laplacian Regularizer

The Laplacian regularizer in Equation (B.19) only uses a sample-based estimator that is unbiased only in the limit that $\sigma \to 0$. To keep the computational overhead at a minimum so that L-MAP can scale to large neural networks, we only take 1 sample of $\psi$ per gradient step. To test the effectiveness of L-MAP in regularizing the eigenvalues of $\mathcal{J}(\theta; p_X)$ under this practical setting, we train an MLP with 2 hidden layers, 16 units, and $\tanh$ activations on the Two Moons dataset (generated with `sklearn.datasets.make_moons(n_samples=200, shuffle=True, noise=0.2, random_state=0)`) for $10^4$ steps with the Adam optimizer and a learning rate of $0.001$. Here we

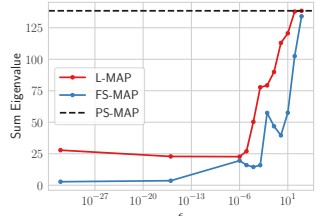

Figure 11: L-MAP is effective at minimizing the eigenvalues.

choose $p_X = \frac{1}{M} \sum_{i=1}^{M} \delta_{x_i}$ with $M = 1,600$ and $\{x_i\}$ linearly spaced in the region $[-5, 5]^2$. In Figure 11, we compare the sum of eigenvalues of $\mathcal{J}(\theta; p_X)$ for L-MAP and FS-MAP with different levels of jitter $\epsilon$. Both FS-MAP and L-MAP significantly reduce the eigenvalues compared to PS-MAP with small values of $\epsilon$, corresponding to stronger regularization.

### C.6 Visualizing the Effect of Laplacian Regualrization

The hyperparameter $\epsilon$ is inversely related to the strength of the Laplacian regularization. We visualize the effect of $\epsilon$ in Figure 12, showing the L-MAP solution varies smoothly with $\epsilon$, evolving from a near-zero function that severely underfits the data to one that fits the data perfectly.

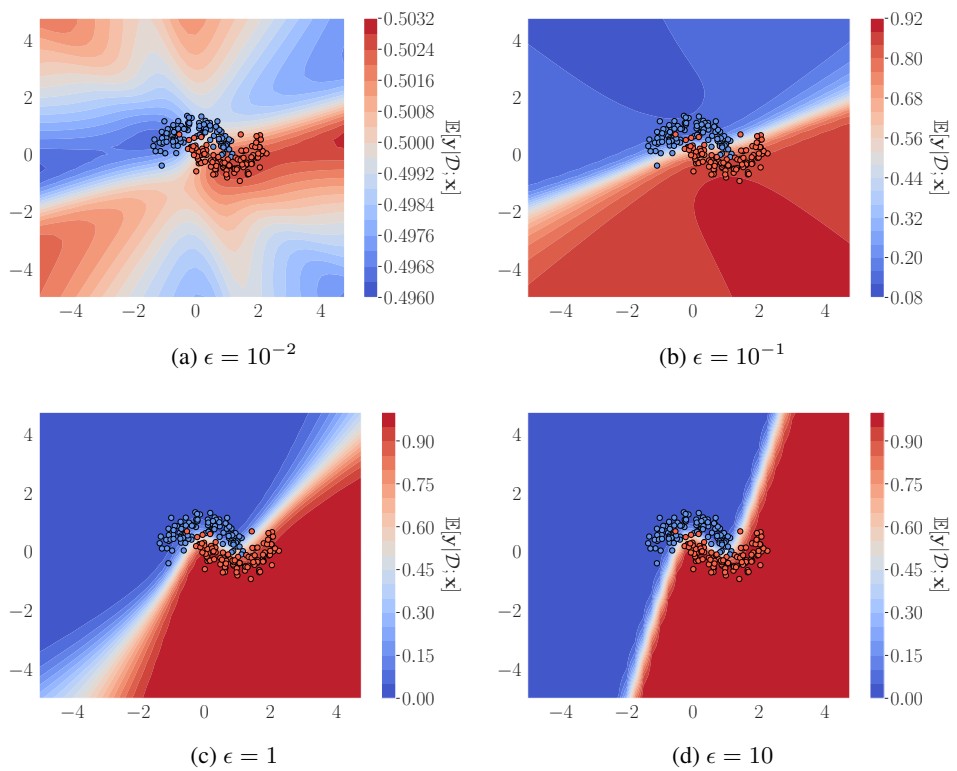

Figure 12: **Visualization of L-MAP solustions at various $\epsilon$.**

## C.7 Further Neural Network Experiments

In Table 4, we report the detailed results for varying the number of Monte Carlo samples $S$ as reported by Section 4.5 and Figure 7. As noted previously, the performance is fairly robust to this hyperparameter $S$ in terms of accuracy, selective accuracy, and negative $\log$-likelihood. However, we find that increasing $S$ can sometimes lead to significant improvement in calibration, as seen for FashionMNIST [31].

Table 4: This table provides detailed quantitative performance for the results plotted in Figure 7, ablating the choice of the number of samples used for evaluation of the Laplacian estimator $S$. The standard deviations are reported over five trials.

| # Samples ($S$) | FashionMNIST | | | | CIFAR-10 | | | |
|---|---|---|---|---|---|---|---|---|
| | Acc. ↑ | Sel. Pred. ↑ | NLL ↓ | ECE ↓ | Acc. ↑ | Sel. Pred. ↑ | NLL ↓ | ECE ↓ |
| 4 | 94.0%±0.1 | 99.2%±0.0 | 0.28±0.01 | 4.3%±0.2 | 95.6%±0.1 | 99.5%±0.0 | 0.17±0.01 | 2.4%±0.1 |
| 8 | 94.0%±0.2 | 99.2%±0.0 | 0.27±0.01 | 4.2%±0.3 | 95.6%±0.1 | 99.5%±0.0 | 0.17±0.00 | 2.2%±0.1 |
| 16 | 94.1%±0.1 | 99.2%±0.0 | 0.27±0.01 | 4.1%±0.1 | 95.6%±0.2 | 99.5%±0.0 | 0.17±0.00 | 2.1%±0.1 |
| 32 | 93.8%±0.1 | 99.2%±0.0 | 0.28±0.01 | 4.4%±0.2 | 95.5%±0.1 | 99.5%±0.0 | 0.16±0.00 | 1.8%±0.2 |
| 64 | 94.0%±0.1 | 99.2%±0.0 | 0.27±0.0 | 4.2%±0.1 | 95.7%±0.2 | 99.5%±0.0 | 0.16±0.00 | 1.7%±0.2 |
| 128 | 94.1%±0.1 | 99.2%±0.1 | 0.26±0.01 | 4.1%±0.1 | 95.5%±0.1 | 99.5%±0.0 | 0.16±0.00 | 1.4%±0.1 |
| 256 | 94.0%±0.2 | 99.2%±0.0 | 0.27±0.0 | 4.3%±0.1 | 95.5%±0.1 | 99.5%±0.0 | 0.16±0.00 | 1.2%±0.2 |
| 512 | 93.9%±0.1 | 99.1%±0.0 | 0.26±0.01 | 4.2%±0.1 | 95.5%±0.1 | 99.5%±0.0 | 0.16±0.00 | 1.2%±0.1 |

**Distribution Shift.** We additionally evaluate our L-MAP trained models to assess their performance under covariate shift. Specifically, for models trained on CIFAR-10 [15], we use the additional test set from CIFAR-10.1 [22] which contains additional images collected after the original data, mimicking a covariate shift. As reported in Table 5, L-MAP tends to improve calibration, while retaining performance in terms of accuracy.

Table 5: We evaluate the performance of CIFAR-10.1 [22] using the models trained on CIFAR-10 [15]. L-MAP tends to improve the data fit in terms of the log likelihood and is better calibrated while retaining the same performance as PS-MAP. The standard deviations are reported over five trials.

| Method | Acc. ↑ | Sel. Pred. ↑ | NLL ↓ | ECE ↓ |
|---|---|---|---|---|
| PS-MAP | 89.2%±0.4 | 97.8%±0.2 | 0.43±0.03 | 6.2%±0.3 |
| L-MAP | 89.2%±0.7 | 97.9%±0.2 | 0.40±0.02 | 4.1%±0.5 |

**Transfer Learning.** Using ResNet-50 trained on ImageNet, we test the effectiveness of L-MAP with transfer learning. We choose hyperparameters as in Appendix C.3, except a lower learning rate of $10^{-3}$ and a smaller batch size of 64 due to computational constraints. Table 3 reports all the results.

