# OpenReview forum: "Should We Learn Most Likely Functions or Parameters?"
_NeurIPS.cc/2023/Conference — NeurIPS 2023 poster_

### Official Review · Reviewer_wSt3 · 2023-06-22

**Soundness:** 3 good
**Presentation:** 4 excellent
**Contribution:** 3 good
**Rating:** 7
**Confidence:** 2

**Summary:**

This paper investigates an important yet neglected question in machine learning: should we train the model in the parameter space or function space? The authors show the benefits and shortage of function space MAP estimation which could provide a guide for practitioners.  Last but not least, the authors propose a scalable approximation for function space MAP estimation for deep learning.

**Strengths:**

- Thorough analysis of the pros and cons of parameter space MAP estimation and function space MAP estimation
- Propose a scalable approximation for large neural networks which extends the application area of function space MAP estimation
- Well written and easy to follow

**Weaknesses:**

- I wonder how universal are the pros and cons demonstrated in the paper, considering they're being shown in a relatively simple toy data set
- I understand the author is not claiming L-MAP works well in all cases, but I am still interested to see the performance of L-MAP on tasks other than classification

**Questions:**

See weakness above

**Limitations:**

See weakness above

---

> ### Author Rebuttal · Authors · 2023-08-10
>
>
> Thank you for your thoughtful and constructive questions and suggestions!
>
> We were pleased that you found our manuscript to be **"well-written and easy to follow"** and that you highlighted our **"thorough analysis"** of the pros and cons of PS-MAP and FS-MAP.
>
> We address your questions and comments below. Please let us know if you have any remaining questions.
>
> ---
>
> > I wonder how universal are the pros and cons demonstrated in the paper, considering they're being shown in a relatively simple toy data set
>
> Thank you for this important question. The goal of our experiments on synthetic datasets was precisely to probe the pros and cons of FS-MAP in a well-controlled setting and the results are highly in line with the expectation from our theoretical analysis. We also conducted extensive evaluations with neural networks on datasets commonly used in deep learning. Moreover, we have run new experiments on UCI regression and transfer learning to broaden our evaluations, which can be found in our general response. Finally, we added a discussion on a comprehensive set of criteria for when FS-MAP and L-MAP should or shouldn't be expected to outperform PS-MAP in more general settings.
>
> ---
>
>
> > I understand the author is not claiming L-MAP works well in all cases, but I am still interested to see the performance of L-MAP on tasks other than classification
>
> We appreciate the suggestions for additional evaluations and ran additional experiments to compare L-MAP and PS-MAP on UCI regression datasets and transfer learning on CIFAR-10.
>
> ### UCI regression
> We found that L-MAP outperformed PS-MAP on 7 out of 8 datasets according to normalized test RMSE, showing L-MAP can also benefit generalization on many regression tasks.
>
> | Dataset   | L-MAP             | PS-MAP            |
> |:----------|:------------------|:------------------|
> | Boston    | 0.352 ± 0.040     | **0.329 ± 0.033** |
> | Concrete  | **0.261 ± 0.013** | 0.272 ± 0.016     |
> | Energy    | **0.041 ± 0.002** | 0.042 ± 0.003     |
> | Naval     | **0.018 ± 0.002** | 0.032 ± 0.005     |
> | Power     | **0.218 ± 0.005** | 0.219 ± 0.006     |
> | Protein   | **0.580 ± 0.005** | 0.584 ± 0.005     |
> | Winered   | **0.792 ± 0.031** | 0.851 ± 0.029     |
> | Winewhite | **0.714 ± 0.017** | 0.758 ± 0.013     |
>
> We adopt the following setup: we use a 3 hidden layer MLP with 256 units and tune the weight decay (prior variance) and Laplacian regularization strengths on a validation set. We standardize both the inputs and targets and report the mean and standard error of test RMSE across six different runs. For L-MAP we set $p_X = \mathcal{N}(0, I)$ since the inputs were standardized and fairly low-dimensional.
>
> ### Transfer Learning from ImageNet to CIFAR-10
>
> Using a ResNet18 pre-trained with ImageNet, we tested the performance on further fine-tuning on CIFAR-10 with samples from CIFAR-100 as the evaluation set for 5 epochs. As with training from scratch, we still use a batch size of 128 and an evaluation points size of 128.
>
> | Method | Acc. | Sel. Acc. | Avg. NLL | ECE |
> | ----- | ----- | ----- | ----- | ----- |
> | PS-MAP | 95.3% | 99.5% | 0.14 | 1.2% |
> | L-MAP | 95.4% | 99.5% | 0.14 | 1.0% |
>
> We find L-MAP is able to perform marginally better while improving the calibration of the classifier slightly.
>
>
> ---
>
> Please let us know if you have any further questions.

---

> > ### Comment · Reviewer_wSt3 · 2023-08-14
> >
> > Thank you for the response, it answered my questions and concerns.

---

> > > ### Author Response · Authors · 2023-08-16
> > >
> > > Thank you for your feedback and for recommending acceptance of our submission! We will take your comments and suggestions into account when updating the manuscript.

---

> > > ### Author Response · Authors · 2023-08-21
> > >
> > > Thank you again for supporting acceptance of our submission! Your feedback and suggestions were very valuable and have helped us further improve our submission, and we will include the additional results and clarifications from our rebuttal in the revised manuscript.
> > >
> > > If you agree that the suggested clarifications and new results improved our submission, we would be grateful if you would consider raising your score to reflect these additions. Thank you for your time and effort!

---

### Official Review · Reviewer_Zoj6 · 2023-06-30

**Soundness:** 4 excellent
**Presentation:** 3 good
**Contribution:** 3 good
**Rating:** 6
**Confidence:** 4

**Summary:**

This work investigates the task of performing inference over model functions rather than model parameters. In particular, the authors propose an objective function *function-space MAP (FS-MAP)*, which intuitively is the usual MAP objective where the prior term is taken over functions rather than parameters. Defining this function-space prior is fundamentally challenging, and to do so, the authors generalize the techniques of Wolpert (1993) to consider functions observed on (possibly) infinite sets.

The authors perform an in-depth empirical evaluation of this proposed objective on several synthetic and real-world benchmarks. In addition, a scalable approximation to the objective is proposed in Section 4.3.

**Strengths:**

- The paper is overall very well-written and clear in its aims and conclusions.
- The key ideas of the paper are likely of interest to the community and the findings have the potential to be impactful for future work on function-space inference.
- I found the discussion relating the function-space geometry to the sampling distribution in Section 3.2 fascinating, and this was a novel connection that I had not considered before.
- The experimental evaluation is generally sound and the conclusions drawn are supported by the evidence provided throughout the paper.
- The proposed objective is a non-trivial extension of previous work in this area, and the scalable approximation in Section 4.3 is similarly non-trivial.

**Weaknesses:**

 - There are several places in the paper lacking rigor and where I believe there to be undressed latent fundamental issues (see the questions section). As noted on Line 96, a fundamental difficulty in function-space inference is defining an appropriate notion of a prior density over infinite dimensional spaces. The methods of this work are obtained via discretization of the functions followed by a heuristic passage to the limit, but it is unclear how this relates to the underlying function-space distributions, which are the true objects of interest.
- The empirical results on real-world data (Table 1) do not show benefits of using the FS-MAP objective over the standard PS-MAP objective.
	- Note that I do not view this necessarily as a weakness of the work itself but rather a potential limitation of the methodology itself, and an honest evaluation of the proposed methodology is valuable.
- The paper needs to be better contextualized within the literature, e.g. there is no related work section. See e.g. [1-4] for some potentially relevant works on function-space inference which are potentially of interest.


[1] Function-Space Regularization in Neural Networks: A Probabilistic Perspective, Rudner et al., ICML 2023

[2] Understanding Variational Inference in Function-Space, Burt et al., AABI 2020

[3] Generalized Variational Inference in Function Spaces: Gaussian Measures meet Bayesian Deep Learning, Wild et al., NeurIPS 2022

[4] Tractable Function-Space Variational Inference in Bayesian Neural Networks, Rudner et al., NeurIPS 2022

**Questions:**

- Under what conditions is Equation (10) well-defined? It was not clear that the integral in this expression is finite without e.g. further assumptions on the model class. (It is straightforward to give examples where the integral is infinite)
- Can the infinite-observation objective (Equation (11)) be justified as an MAP estimate? Equation (8) follows from the change-of-variables formula, but it was not clear to me if (or how) Equation 11 was being justified in the same manner, or merely as a heuristic.

**Limitations:**

The authors appropriately discuss the limitations of their work within the submission.

---

> ### Author Rebuttal · Authors · 2023-08-10
>
> Thank you for your thoughtful and constructive questions and suggestions!
>
> We were pleased that find that our submission is **"of interest to the community"** and that our findings **"have the potential to be impactful for future work on function-space inference"**. We were also happy to see that you found the manuscript to be **"very well-written"** and **"clear in its aims and conclusions"**.
>
>
> We address your questions and comments below. Please let us know if you have any remaining questions.
>
> ---
> > [...] The methods of this work are obtained via discretization of the functions followed by a heuristic passage to the limit, but it is unclear how this relates to the underlying function-space distributions
>
> You correctly noted that the main difficulty in establishing any connection between our result and the so-called true object of interest involving the density evaluation $p(f_\theta)$ is due to technical challenges in defining the infinite-dimensional probability density function $p(f_\theta)$ in the first place.
>
> Section 3.2 establishes the limiting behavior of the finite-point objective derived by Wolpert (which exactly corresponds to the value of the prior probability density function of a distribution over finite function evaluations computed at a given parameterized function $f_\theta$) and thereby avoids explicitly defining $p(f_\theta)$. We expect the evaluation of $p(f_\theta)$ obtained using the infinite-limit expression to agree with an evaluation that could be obtained using a suitably-defined base measure over the function space. A justification for this is provided in Appendix B.2. The technique used in Section 3.2 is referred to as taking the continuum limit and has previously been applied to analogous problems in physics and applied mathematics.
>
> For example, in numerical analysis, the solutions of discretized differential equations often converge to the solutions of the continuous equations as the mesh size goes to zero. Similarly, lattice field theories in physics, where the continuous spacetime is discretized into a lattice, has provided consistent results with the continuum approach in many cases.
>
> ---
> > The empirical results on real-world data (Table 1) do not show benefits of using the FS-MAP objective over the standard PS-MAP objective.
>
> Thank you for pointing out that further discussion of the results would be useful. Indeed, as we discussed in our general response, we were not expecting FS-MAP or L-MAP to lead to significant improvements in these settings where we do not have a highly informative prior. In contrast, we use a prior that exactly corresponds to the data-generating process in Section 3.3, leading to significant gains by performing FS-MAP. Here, since our Gaussian prior is rather non-informative about the data-generating process in the FashionMNIST and CIFAR-10 experiments, we were pleasantly surprised to find that L-MAP achieves a significant reduction in ECE on CIFAR-10 with $p\_X=$CIFAR-100 and noticeable improvement in selective prediction accuracy on FashionMNIST with all three choices for $p\_X$, though the accuracy is comparable with PS-MAP.
>
> ---
> > Under what conditions is Equation (10) well-defined? It was not clear that the integral in this expression is finite without e.g. further assumptions on the model class.
>
> It is true that Equation (10) is not finite for any model. However, for most models, including commonly used neural networks, we can show it is indeed finite. To show the expectation in Equation (10) is finite, it is sufficient to show that the integrand $J_\theta(x)^\top J_\theta(x)$ is bounded for any $x$ in the support of $p_X,$ a sufficient condition for which is that the function $f_\theta$ has bounded Lipschitz constants over the support of $p_X$. Consequently, Equation (10) is finite for any $\theta$ for a wide range of models, e.g., any MLP with standard activation functions (ReLU, tanh). Note this is not a necessary condition, and Equation (10) can still be finite depending on the specifics of $p_X$ and $J_\theta(x).$
>
> ---
> > Can the infinite-observation objective (Equation (11)) be justified as an MAP estimate? [...] it was not clear to me if (or how) Equation 11 was being justified in the same manner, or merely as a heuristic.
>
> As mentioned above, the main difficulty in establishing any connection between our result and the true mode of $p(f_\theta | \mathcal{D})$ lies in technical challenges in the prior density $p(f_\theta)$ in the first place. On the other hand, our result by taking the continuum limit is not merely a heuristic but follows an established approach with a history of successful applications in other areas of mathematics and physics that enable tractable computational methods when the continuum approach presents challenges.
>
> ---
> > The paper needs to be better contextualized within the literature, e.g. there is no related work section.
>
> Function-space regularization in NNs is a burgeoning field, and we appreciate the opportunity to further contextualize our work:
>
> [1] discuss function-space VI in NNs. They explain in which cases the function-space variational objective proposed in [2] is not well-defined. This pathology is conceptually similar to the pathology identified in our work.
>
> [3] propose approximations to make the variational objective proposed by [2] tractable and identify a prior over functions for which the objective is well-defined. Unlike our approach, the sampling distribution in [3] is not part of the probabilistic model but of the approximation.
>
> [4] also perform variational inference but they minimize a Wasserstein instead of a KL divergence, and they consider Gaussian process models instead of stochastic NNs.
>
> [5] specify a prior over parameters that induce a desirable prior over functions but perform inference in parameter space instead of deriving a function-space objective to avoid the challenges described in our submission.
>
> ---
> Please let us know if you have any further questions.

---

> ### Author Response · Authors · 2023-08-21
>
> Thank you again for supporting acceptance of our submission! Your feedback and suggestions were very valuable and have helped us further improve our submission, and we will include the additional results and clarifications from our rebuttal in the revised manuscript.
>
> If you agree that the suggested clarifications and new results improved our submission, we would be grateful if you would consider raising your score to reflect these additions. Thank you for your time and effort!

---

### Official Review · Reviewer_PZwV · 2023-07-07

**Soundness:** 3 good
**Presentation:** 3 good
**Contribution:** 2 fair
**Rating:** 6
**Confidence:** 2

**Summary:**

This paper analyzes the question: should we find the parameter that maximizes $p(\theta|D)$ or $p(f_\theta|D)$, given a data distribution $D$.
The former is the classic MAP, or PS-MAP ("parameter space") in this work, and the latter is called FS-MAP ("function space").

The paper shows that neither is universally more desirable than the other, and describes pros and cons of both methods:
- FS-MAP has the advantage of directly optimizing the quantity we care about (i.e. function value), is invariant to reparameterization, and can be more robust to label noise.
- On the other hand, PS-MAP tends to be more stable, and while it is not invariant to reparameterization, it's unclear how much this would be a disadvantage in practice. Moreover, PS-MAP can be closer to the Bayesian model average than FS-MAP.

The findings are supported by both theory and experiments.

**Strengths:**

- The paper has extensive comparisons on the limitations and applicability of both methods.
- The paper addresses practical considerations:
  - The problem of having a (near) singular Jacobian is addressed by adding perturbations.
  - For scalability, the problem of requiring two forward steps is addressed by zeroth-order approximation, which requires 1 forward pass only.


**Weaknesses:**

While I appreciate the discussions, I'm not sure the evidences are clear enough.
- Fig 2 shows that there's no difference between FS-MAP and PS-MAP when the number of samples is around 10k, a sample size that most practical settings can afford.
- In Table 1, the results from both methods are very close, often within the margin of standard deviation.

Overall, it seems that both methods compare similarly, and it's unclear how much we would lose by simply always choosing PS-MAP.

**Questions:**

- Sec 3.3: the choice of prior is important; is there guidance on how to choose a prior in practice?
- Is there guidance on how to choose between FS-MAP and PS-MAP given a practical problem? How much would the choice matter (the experiments seem to show that the choice doesn't matter much)?
- Fig 2b: why is there an increase in curvature a the number of samples increases?
    - also, is there a reason for measuring flatness as the average eigenvalue (rather than e.g. max)?


**Limitations:**

The paper clearly discusses the limitations.

There is no direct societal impact.

---

> ### Author Rebuttal · Authors · 2023-08-10
>
> Thank you for your thoughtful and constructive questions and suggestions!
>
> We were pleased that you highlighted the **"extensive comparison"** of limitations and applicability of PS-MAP and FS-MAP and that the paper addresses **"practical considerations"**.
>
> We address your questions and comments below. Please let us know if you have any remaining questions.
>
> ---
>
> > In Table 1, the results from both methods are very close, often within the margin of standard deviation.
>
> Thank you for pointing out that further discussion of the results would be useful. Indeed, as we discussed in our general response, we were not expecting FS-MAP or L-MAP to lead to significant improvements in these settings where we do not have a highly informative prior. In contrast, we use a prior that exactly corresponds to the data-generating process in Section 3.3, leading to significant gains by performing FS-MAP.  Here, since our Gaussian prior is rather non-informative about the data-generating process in the FashionMNIST and CIFAR-10 experiments, we were pleasantly surprised to find that L-MAP achieves significant reduction in ECE on CIFAR-10 with $p\_X=$CIFAR-100 and noticeable improvement in selective prediction accuracy on FashionMNIST with all three choices for $p\_X$, though the accuracy is comparable with PS-MAP.
>
> ---
>
> > Fig 2 shows that there's no difference between FS-MAP and PS-MAP when the number of samples is around 10k, a sample size that most practical settings can afford.
>
> Thank you for the observation. This early convergence to low test error for both methods is because the data is generated exactly from our prior, greatly simplifying the task and reducing the sample complexity to a much lower level compared to many real-world applications. Relative to the complexity of this task, FS-MAP still achieves a noticeable improvement in efficiency compared to PS-MAP.
>
> ---
>
> > Sec 3.3: the choice of prior is important; is there guidance on how to choose a prior in practice?
>
> Indeed, Sec 3.3 shows that well-specification of the prior as well as $p_X$ was important to achieving the best performance with FS-MAP. Choosing a good prior often involves exploiting problem-specific inductive biases (e.g., convolutional networks induce a prior that favors translation equivariant functions, suitable for problems in computer vision) and is a fundamental question in machine learning and outside the scope of this work. On the other hand, our work does provide guidance on choosing $p_X$ by revealing its connection to the metric in function space (Section 3.2) and shows choosing $p_X$ to closely approximate the input distribution at test time often works better (Figure 3).
>
> ---
>
> > Is there guidance on how to choose between FS-MAP and PS-MAP given a practical problem? How much would the choice matter (the experiments seem to show that the choice doesn't matter much)?
>
> In general, despite the flatness-favoring properties of FS-MAP, there is no strong reason to expect that either FS-MAP or PS-MAP will generalize better in the absence of a good prior that is highly informative of the data-generating process. When such a prior is available, we expect that the choice between FS-MAP and PS-MAP to be more important and FS-MAP to more likely outperform PS-MAP. We provide a comprehensive discussion on potentially useful criteria we can rely on to choose between FS-MAP or PS-MAP in the general response.
>
> ---
>
>
> > Fig 2b: why is there an increase in curvature a the number of samples increases?
>
> Thanks for the intriguing observation! There is indeed a slight increase in curvature for FS-MAP as we increase training samples. We hypothesize this is because as we add more training data, the volume of solutions that can fit the data decreases since more constraints need to be satisfied. As a result, it becomes more difficult to not increase the loss as we move away from the minimum.
>
> ---
>
> > is there a reason for measuring flatness as the average eigenvalue (rather than e.g. max)?
>
> We didn't have any strong reason against using max, though taking the mean captures more information about the local geometry as it is aggregated over all dimensions of the high dimensional parameter space, whereas the max eigenvalue only measures the loss curvature in a single direction.
>
>
> ---
>
> Please let us know if you have any further questions!

---

> > ### Comment · Reviewer_PZwV · 2023-08-18
> >
> > Thank you very much for the detailed response and for the additional empirical results!
> >
> > I still think that the paper can be further strengthened if the practical takeaway is clearer (since the comparisons are close and mostly at a small scale), but I think the current results are interesting enough for an acceptance, so I'm raising my score.

---

> > > ### Author Response · Authors · 2023-08-18
> > >
> > > Thank you for your feedback and for recommending acceptance of our submission! We will take your comments and suggestions into account when updating the manuscript.

---

### Official Review · Reviewer_VTcx · 2023-07-21

**Soundness:** 3 good
**Presentation:** 3 good
**Contribution:** 3 good
**Rating:** 7
**Confidence:** 2

**Summary:**

The paper compares the widely-used parameter estimation of machine learning against function estimation for the maximum a posterior (MAP) estimation. The authors provide detailed analysis and mathematical variation why there are significant difference in results for function vs. paramter MAP, and introduce conditions where the function MAP can avoid its failure modes.

To be honest, I am not an expert in the area, and cannot make an assessment how novel the proposed method is against relevant literature. I would rely on other reviewers on the aspect of the evaluation.

**Strengths:**

- The paper is fairly well-written in most of cases, and grounds strong motivation and intuition for the proposed idea. Even though I am not expert in the area, I enjoyed reading through the paper.



**Weaknesses:**

- The paper is heavy on math. Even though they authors did a decent job to make the equations approachable, some of important definitions are missing, and many of details are deferred to the supplementary material. It is suggested to double-check the completeness of the formulation in the main paper. If it cannot be self-contained, I am afraid the publication should head to another venue with a longer length limit (maybe a journal). For example, P was never introduced, but could be only inferred as the parameter dimension. Also, $\theta_R$ and $\theta_L$ was not clearly defined in Figure 1, which is not desirable considering that it should serve as a strong motivating example.

- The results are not very competitive. Especially for Table 1 for MNIST and CIFAR-10. Is there any intuitions why it is the case? On the other hand, graphs in Figures appear quite promissing. I would appreciate more discussion in the paragraph on benchmark performance (line 297-298).

**Questions:**

- This might be simply my ignorance. But what is the difference between N samples ($x_\mathcal{D}^{(i)}$) vs. M samples ($\hat{x}$) used in Equation 5? Is it necessary to derive different samples from the space, or can we use the same samples? Roughly how big number would you choose for N and M, respectively? The paper writes condition based on M ($MK \geq P$). But is there any condition on N?

**Limitations:**

The authors clearly stated limitations in extensive aspects. However, it would be informative to have a detailed numbers on the increased numerical complexity or time/ memory requirement for using FS-MAP (or LS-MAP) over PS-MAP for some of famous architecture.

---

> ### Author Rebuttal · Authors · 2023-08-10
>
>
> Thank you for your thoughtful and constructive questions and suggestions!
>
> We were pleased that you **"enjoyed reading the paper"** and that you found it to be **"well-written"**.
>
> We address your questions and comments below. Please let us know if you have any remaining questions.
>
> ---
>
> > Even though they authors did a decent job to make the equations approachable, some of important definitions are missing. [...] For example, P was never introduced, but could be only inferred as the parameter dimension. Also, $\theta_R$ and $\theta_L$ was not clearly defined in Figure 1, which is not desirable considering that it should serve as a strong motivating example.
>
> Thank you for suggesting clearer definitions of notations. We will clarify that $P$ is the number of parameters and $\theta \in \mathbb{R}^P$ in the preliminaries. We will also clarify that $\theta\_L$ and $\theta\_R$ refers to the parameters controlling the height of the left and right Gaussian, respectively. If you think it would be helpful, we will include a list of definitions in the appendix.
>
>
> ---
>
>
> > The results are not very competitive. Especially for Table 1 for MNIST and CIFAR-10. Is there any intuitions why it is the case? On the other hand, graphs in Figures appear quite promising. I would appreciate more discussion in the paragraph on benchmark performance (line 297-298).
>
> Thank you for pointing out that further discussion of the results would be useful. We included such a discussion in our general response and will add it to the manuscript. As discussed in our general response, we were not expecting FS-MAP or L-MAP to lead to significant improvements in these settings where we do not have a highly informative prior. In contrast, we use a prior that exactly corresponds to the data-generating process in Section 3.3, leading to significant gains by performing FS-MAP.  Here, since our Gaussian prior is rather non-informative about the data-generating process in the FashionMNIST and CIFAR-10 experiments, we were pleasantly surprised to find that L-MAP achieves significant reduction in ECE on CIFAR-10 with $p\_X=$CIFAR-100 and noticeable improvement in selective prediction accuracy on FashionMNIST with all three choices for $p\_X$, though the accuracy is comparable with PS-MAP.
>
>
>
> ---
>
>
> > This might be simply my ignorance. But what is the difference between $\mathrm{N}$ samples $(x_{\mathcal{D}}^{(\imath)})$ vs. $M$ samples $(\hat{x})$ used in Equation 5? Is it necessary to derive different samples from the space, or can we use the same samples? Roughly how big number would you choose for $N$ and $M$, respectively? The paper writes conditions based on $M$ ($MK > P$). But is there any condition on N?
>
> $N$ is the number of points in the training set $x_{\mathcal{D}}$ and there is no condition based on $N$. $M$ is the number of points in the set of evaluation points $\hat{x}$ for computing the prior $p(f\_{\theta}(\hat{x}))$. $\hat{x}$ and $x_{\mathcal{D}}$ ar generally not the same since $\hat{x}$, being a special case of specifying an evaluation distribution $p_X,$ corresponds to a choice of metric in function space as we show in Section 3.2. Technically, using the same samples is possible, and we investigate setting $\hat{x} = x\_{D}$ in our experiments (see $p\_X = Train$ in Table 1). More generally, however, we wish to set $M$ to be as large as possible to appropriately account for the behavior of the function throughout the entire input space, and Figure 3a shows that FS-MAP indeed can benefit from using larger $M.$ While it is generally intractable to use exceedingly large $M$ as discussed in Section 4.3, there we also present an efficient approximation via L-MAP that doesn't suffer from this intractability by allowing unbiased Monte-Carlo estimates for objectives defined with arbitrarily large $M$.
>
>
> ---
>
> > The authors clearly stated limitations in extensive aspects. However, it would be informative to have a detailed numbers on the increased numerical complexity or time/ memory requirement for using FS-MAP (or LS-MAP) over PS-MAP for some of famous architecture.
>
> Compared to PS-MAP, L-MAP only requires an additional forward pass on $S$ evaluation points to evaluate the Laplacian regularization term, where $S$ is the number of Monte Carlo samples. This is much faster compared to the exact FS-MAP, as we discuss in Section 4.3 line 291-293.
>
> To showcase concrete run times, we train an MLP for 10,000 steps on the Two Moons datsets with a batch size of $200$ and $1600$ evaluation points. The MLP has only 2 hidden layers and 16 units each so that FS-MAP is tractable. L-MAP, finishing in 31 seconds, is 33 times faster than FS-MAP, which took 16 minutes, and only 1.4 times slower than PS-MAP, which took 22 seconds. We also show the time per gradient step and peak memory consumption for ResNet-18:
>
>
> For the scalable experiments with neural networks in Section 4.3, we report the approximate wall clock times between standard PS-MAP and our proposed approximation L-MAP in the table below. All experiments were run with batch size $128$ and $S=128$ Monte Carlo samples for the evaluation points. As expected, L-MAP takes approximately twice the amount of time due to the additional forward pass. Note it is not feasible to run FS-MAP in this case as it requires too much memory.
>
>
> | Dataset | Method | Gradient Step (ms) | Epoch (s) |
> | ------- | -------- | -------- | -------- |
> | FashionMNIST | PS-MAP     |  54     |  24     |
> |  | L-MAP     |  109     |  45    |
> | CIFAR-10 | PS-MAP     |  64     |  27     |
> |  | L-MAP     |  126     |  52    |
>
> All times are reported based on an NVIDIA TITAN RTX 24 GB GPU, where each run takes approximately 7GB of GPU memory in JAX.
>
>
> ---
>
> Please let us know if you have any further questions!

---

> > ### Comment · Reviewer_VTcx · 2023-08-14
> >
> > Thanks for the reply. It cleared most of my concerns.

---

> > > ### Author Response · Authors · 2023-08-16
> > >
> > > Thank you for your feedback and for recommending acceptance of our submission! We will take your comments and suggestions into account when updating the manuscript.

---

> > > ### Author Response · Authors · 2023-08-21
> > >
> > > Thank you again for supporting acceptance of our submission! Your feedback and suggestions were very valuable and have helped us further improve our submission, and we will include the additional results and clarifications from our rebuttal in the revised manuscript.
> > >
> > > If you agree that the suggested clarifications and new results improved our submission, we would be grateful if you would consider raising your score to reflect these additions. Thank you for your time and effort!

---

### Official Review · Reviewer_NXLr · 2023-07-25

**Soundness:** 3 good
**Presentation:** 3 good
**Contribution:** 3 good
**Rating:** 5
**Confidence:** 3

**Summary:**

This paper explores the question of learning most likely functions vs most likely parameters in the context of MAP estimation, a common setting including minimizing log likelihood with L1 or L2 regularization, as these regularizations correspond to imposing a prior. Differences in parameterization affect the MAP through change of measure for the prior, and potentially in very unhelpful ways. Because of these potential problems with parameterizations, authors explore doing MAP not on parameters as we usually do (referred to as PS-MAP), but on function outputs for a set of inputs, an idea from Wolpert (1993). Authors refer to this as function-space MAP (FS-MAP).

The contributions are as follows:
- While Wolpert considered function outputs on a finite set of inputs, authors generalize to function outputs on a measure on input space. Authors emphasize that this measure of input space can be specified according to the task at hand (e.g. restricting to natural images, rather than all images).
- Authors empirically compare FS-MAP and PS-MAP on a simple example and empirically find that FS-MAP finds flatter minima, has better generalization in some settings, and produces less overfitting. Authors also discuss how FS-MAP and PS-MAP compare with the posterior mean in variations on this simple setting. Also, FS-MAP’s test performance depends on the measure of inputs that are considered important.
- Authors discuss problems with FS-MAP: FS-MAP can be computationally very expensive, and can behave in pathological ways under certain conditions. Authors also provide conditions under which it is well-behaved, as well as a scalable approximation to FS-MAP, called L-MAP, that also behaves ok even when FS-MAP would behave pathologically. Authors empirically compare FS-MAP and PS-MAP on neural networks for image classification. L-MAP performs similarly to PS-MAP, but is better calibrated in one experimental setting, and also finds flatter minima.

**Strengths:**

1. Authors raise and discuss an interesting question that I had not thought about.
2. There is a substantive discussion of PS-MAP and FS-MAP, including weaknesses of both approaches.
3. Authors provide a computationally feasible, well-behaved approximation to a computationally too-expensive and poorly-behaved objective (L-MAP, vs FS-MAP).

**Weaknesses:**

1. There is a tension between statements like “parameters have no meaning independent of the functional form of the models they parameterize” (lines 22-23) and the fact that the difference between best parameters vs best functions is a matter of parameterization of the prior, and the prior is a prior on the parameters (Equation 1). This tension is central to the question of parameterization for MAP and is not addressed.
2. This tension continues into the experiments, many of which emphasize flat minima. Authors cite Dinh’s “Sharp minima can generalize for deep nets”, which points out that the flatness / sharpness of minima is a function of parameterization.
3. Function-space MAP and priors on functions are defined much later in the paper than ideal. For example, in Figure 1, I do not know how FS-MAP is calculated. p_X (important to FS-MAP) is also not specified.
4. The terminology of “most likely functions” and “more probable functions” obscures the dependence of these terms on the choice of p_X (e.g. line 37, 184).
5. The connection to variational inference does not seem particularly strong. MAP as VI would involve choosing the best q, while the construction in Section B.9 does not do that. It’s also unclear to me why we care about these constructions; e.g. why do we care about q, besides that it ultimately leads to (13) and (B.13)?
6. Also, (B.8) looks like a KL divergence, not a variational lower bound / ELBO. Maybe I’m missing something here.
7. The experiments and discussion around experiments could be more thorough. See questions.
8. If someone told me they were looking at most likely functions, rather than most likely parameters for MAP, I would be curious about priors on functions in place of priors on parameters. (This isn't a weakness / question / limitation; it's more of a comment.)

**Questions:**

1. On line 135-136, “the distance between two function evaluations is measured locally by [...]” : what two function evaluations are being referred to?
2. Why is it important to have infinitely many evaluation points? I assume that’s not the appeal but that’s how it was advertised. (Lines 127-128)
3. What is p_X=N/A in Figure 3b?
4. Can you provide intuition for why FS-MAP archives better generalization, in the settings where it does achieve better generalization? Can you also provide intuition for why it is less prone to overfitting? Can you discuss when this wouldn’t be the case? (Section 3.3)
5. The numbers in Table 1 are all very similar. What do you think is the reason for this? Should we be surprised by the similarities between the different choices of p_X, given how much p_X mattered in Section 3.3?
6. “For example, while it is 331 natural to assume that FS-MAP more closely approximates a Bayesian model average”: Why is it natural to make this assumption? (lines 330-331)
7. Could you briefly discuss why the posterior mean is particularly important (line 254)? The citation that follows is a textbook.
8. Why do we want to constrain eigenvalues of the Jacobian? I understand that you can run into problems if they are quite large. Is that the only reason? Is there a risk of over-constraining?
9. Why is calibration better with FS-MAP for CIFAR-10 and p_X=CIFAR-100 compared to PS-MAP, but not other settings?
10. Can you explain further why L-MAP is more robust to training with label noise? The explanation in line 325 is Section 3.3, which appears to be about input space, not about label noise
11. Should I expect these comparisons between FS-MAP and PS-MAP in other settings? In what settings should these results hold vs not (beyond satisfying the assumptions)?
12. While Wolpert’s evaluation on finite sets of points is a special case of the authors’ formulation, are they not effectively the same in practical settings?
13. Authors discuss a measure on which to consider function outputs. I am curious about the connections to distribution shift.

**Limitations:**

The experiments and discussion around experiments could be more thorough. See questions. It's not clear to me which experimental results should generalize, and under what conditions.

---

> ### Author Rebuttal · Authors · 2023-08-10
>
> We're glad you appreciated the paper and found it well-written. We address your comments below.
>
> Please let us know if you have any remaining questions.
>
> 1. **When and why FS-MAP can outperform PS-MAP and reduce over-fitting**
> Thank you for asking these important questions at the heart of this paper. While in general there is little reason to expect either FS-MAP or PS-MAP will perform better, we provide a thorough discussion on important special cases in the general response that explains our empirical findings. In terms of reducing overfitting, a simple reason why FS-MAP can reduce overfitting is the introduction of the additional regularization term which can prevent the model from fitting the data. In addition, fitting noise often requires precise settings of certain parameters to account for unnatural noisy patterns, which would incur a large Jacobian determinant penalty.
>
> 2. **Infinitely many evaluation points**
> When using only a finite set of evaluation points $\hat{x}$ as in Section 3.1, one can only find the MAP estimate for the function evaluated at a finite set of points $f\_\theta(\hat{x})$, which is different from the MAP estimate for the function $f\_\theta(\cdot),$ equivalent to an infinite dimensional vector. As our goal is to find the MAP estimate for the whole function $f\_\theta(\cdot),$ we seek to use an infinite set of evaluation points. In Figure 3a, we also empirically show that FS-MAP performs monotonically better as we use more evaluation points and achieves optimal performance with an infinite set.
>
> 3. **Distinction from Wolpert's formulation**
> Our formulation is different from Wolpert's formulation even when using finite samples, since a random finite set of points is sampled at every gradient step and the underlying $p\_X$ can have support on infinitely many points.
>
> 4. **Benchmark Performance**:
> Our prior in FashionMNIST and CIFAR-10 experiments was non-informative, as a result we do not expect FS-MAP / L-MAP to outperform PS-MAP though it did improve calibration in many cases. In contrast, the improvements in Section 3.3 stemmed from using a prior matching the data-generating process. We additionally evaluate L-MAP on UCI dataset and transfer learning on CIFAR-10 and present the results in the the general response.
>
> 5. **Dependence of mostly likely functions on $p_X$**
> We agree that this is an important subtlety. Indeed, revealing this subtlety and establishing the connection to choosing a metric in function space is a key contribution of this work and we will emphasize it more. We additionally note that such dependence on an underlying metric applies to any MAP estimation problems in continuous spaces, including PS-MAP, as the probability density always depends on a choice of base measure.
>
> 5. **FS-MAP approximating BMA**
> It is natural to assume FS-MAP approximates BMA better due to arguments we presented in line 259-263. Namely, with sufficient amount of data, both $p(\theta | \mathcal{D})$ and $p(f_\theta | \mathcal{D})$ are approximately Gaussian. Given Gaussian posteriors, the BMA function coincides with its posterior mode $f_{\theta}$, i.e., FS-MAP. But since PS-MAP seeks the posterior mode of $\theta$ rather than $f_\theta,$ it learns a different function compared to FS-MAP and thus BMA.
>
> 6. **Posterior mean importance**:
> The posterior predictive mean function combines all possible model hypotheses weighted by their posterior likelihood, rather than a single point estimate. It often achieves better generalization and uncertainty estimates as a result [1].
>
> 7. **On model parameterization**
> We emphasize that given any prior $p(\theta)$ and parameterization $f_\theta$, PS-MAP will learn different functions upon reparameterization as we show in Appendix A, whereas FS-MAP will always learn the same function. So while the precise difference between PS-MAP and FS-MAP depends on PS-MAP's parameterization, the existence of this difference is universal. Similarly, while flatness metrics such as Hessian eigenvalues are not parameterization-invariant, FS-MAP's preference for flat minima is universal when compared with PS-MAP as evident from Equation 11. As such, we believe there is no tension between these observations but would appreciate further clarifications.
>
> 8. **On the L-MAP derivation**
> Thank you for catching our typo. Eqn. (B.8) is indeed a **negative** evidence lower bound. Regarding motivation for considering variational inference (VI), unlike in conventional VI, the goal of the formulation described in Appendix B.9 is not primarily to learn an accurate approximation to the posterior but to obtain a non-pathological objective in a principled way. In addition, we do choose the best $q$ within the variational family defined in Appendix B.9 through learning the optimal mean parameter $\theta$.
>
> 9. **Numerical Complexity**:
> L-MAP requires an additional forward pass on $ S $ evaluation points. It's faster than FS-MAP and only slightly slower than PS-MAP. For a Two Moons dataset experiment with an MLP, L-MAP was 33 times faster than FS-MAP and only 1.4 times slower than PS-MAP. In scalable experiments, L-MAP takes roughly twice the time of PS-MAP.
>
> | Dataset | Method | Gradient Step (ms) | Epoch (s) |
> | ------- | -------- | -------- | -------- |
> | FashionMNIST | PS-MAP     |  54     |  24     |
> |  | L-MAP     |  109     |  45    |
> | CIFAR-10 | PS-MAP     |  64     |  27     |
> |  | L-MAP     |  126     |  52    |
>
> 11. **Calibration**:
> We believe L-MAP's calibration improvement on CIFAR-10 with $ p_X = $ CIFAR-100 stem from the evaluation distribution covering relevant regions of the input space without duplicating the training set.
>
> 12. **Notation**:
> We'll clarify unclear notations. Specifically, $ p_X = N/A $ corresponds to PS-MAP in Figure 3b and in line 135-136 the the functions are $f$ and $f+df$ where $df$ is an infinitesimal separation.
>
>
>
> [1] Bayesian deep learning and a probabilistic perspective of generalization, Wilson et al., NeurIPS 2020

---

> > ### Comment · Reviewer_NXLr · 2023-08-14
> >
> > Thanks for your response; it addresses most of my concerns and I am also now better informed.
> >
> > I see that there is much discussion about the performance of FS-MAP vs PS-MAP, but not of how to make sense of the question of "best functions" vs "best parameters" given that the setting has a prior on the *parameters*. Perhaps this is not the point of the paper, but I am still wondering.

---

> > > ### Author Response · Authors · 2023-08-16
> > >
> > > Thank you! We are glad to hear your concerns are addressed.
> > >
> > > While the details of the distinction between most likely functions and parameters does depend on the model parameterization, the existence of this distinction is generic, regardless of the parameterization. Moreover, considering a  prior on parameters does not limit the scope of our work. Indeed, any prior in function space can be equivalently expressed as a prior in parameter space using the identity in Equation 6 (or its generalization we presented in Section 3.2).

---

> > > ### Author Response · Authors · 2023-08-21
> > >
> > > Thank you again for supporting acceptance of our submission! Your feedback and suggestions were very valuable and have helped us further improve our submission, and we will include the additional results and clarifications from our rebuttal in the revised manuscript.
> > >
> > > If you agree that the suggested clarifications and new results improved our submission, we would be grateful if you would consider raising your score to reflect these additions. Thank you for your time and effort!

---

### Author Rebuttal · Authors · 2023-08-10

## General Response to All Reviewers

We thank all reviewers for their feedback, support, and unanimous recognition that our paper is interesting, well-written, and thorough in presenting the pros and cons of the considered approaches. We start by providing a general response aimed at addressing the most common questions from the reviewers.

We first present additional experiments and find that L-MAP outperforms PS-MAP on UCI Regression and transfer learning from ImageNet to CIFAR-10 and then present a comprehensive set of criteria on when FS-MAP and L-MAP should or shouldn't be expected to outperform PS-MAP in general settings, explaining why FS-MAP significantly outperforms PS-MAP in some of our experiments but no others. We hope our response addresses the key points and inquiries raised by the reviewers and can be taken into account in the final assessment.

## Additional Experiments on UCI Regression and Transfer Learning

We provide additional regression and transfer learning experiments with L-MAP, as suggested by some reviewers. The results are consistent with the results in the manuscript and show that L-MAP tends to perform as well or better than PS-MAP.

### UCI

We found that L-MAP outperformed PS-MAP on 7 out of 8 datasets according to normalized test RMSE, showing L-MAP can also benefit generalization on many regression tasks.

| Dataset   | L-MAP             | PS-MAP            |
|:----------|:------------------|:------------------|
| Boston    | 0.352 ± 0.040     | **0.329 ± 0.033** |
| Concrete  | **0.261 ± 0.013** | 0.272 ± 0.016     |
| Energy    | **0.041 ± 0.002** | 0.042 ± 0.003     |
| Naval     | **0.018 ± 0.002** | 0.032 ± 0.005     |
| Power     | **0.218 ± 0.005** | 0.219 ± 0.006     |
| Protein   | **0.580 ± 0.005** | 0.584 ± 0.005     |
| Winered   | **0.792 ± 0.031** | 0.851 ± 0.029     |
| Winewhite | **0.714 ± 0.017** | 0.758 ± 0.013     |

We adopt the following setup: we use a 3 hidden layer MLP with 256 units and tune the weight decay (prior variance) and Laplacian regularization strengths on a validation set. We standardize both the inputs and targets and report the mean and standard error of test RMSE across six different runs. For L-MAP we set $p_X = \mathcal{N}(0, I)$ since the inputs were standardized and fairly low-dimensional.

### Transfer Learning from ImageNet to CIFAR-10

Using a ResNet18 pre-trained with ImageNet, we tested the performance on further fine-tuning on CIFAR-10 with samples from CIFAR-100 as the evaluation set for five epochs. As with training from scratch, we still use a batch size of 128 and an evaluation points size of 128.

| Method | Acc. | Sel. Acc. | Avg. NLL | ECE |
| ----- | ----- | ----- | ----- | ----- |
| PS-MAP | 95.3% | 99.5% | 0.14 | 1.2% |
| L-MAP | 95.4% | 99.5% | 0.14 | 1.0% |

We find L-MAP is able to perform marginally better while improving the calibration of the classifier slightly.

## When FS-MAP should or shouldn't outperform PS-MAP

Many reviewers noted that while FS-MAP leads to noticeably better performance in the synthetic example in Section 3.3, less improvement is observed in neural network experiments on image classification in Section 4.4. We note that these results are exactly in line with our theoretical observation and empirical findings in Section 3.3 that FS-MAP's superior performance depends on a well-specified probabilistic model (prior and likelihood). As one of our key points in the paper, in general, there is no strong reason to expect either FS-MAP or PS-MAP will generalize better since our prior can be arbitrarily different from the true data-generating process. Nevertheless, we now summarize our current best understanding of when FS-MAP should or shouldn't outperform PS-MAP.

The example in Section 3.3 exemplifies four criteria that generally favor FS-MAP:

1) The Jacobian is non-singular everywhere and therefore, FS-MAP does not lead to pathological solutions as described in Section 4.1.
2) The likelihood and the prior exactly correspond to the data-generating process. Consequently, decision theory states that Bayesian Model Average (BMA) is the optimal predictor in terms of expected test RMSE.
3) The prior $p(\theta) = \mathcal{N}(\theta | 0, 10^2)$ is diffuse in parameter space but fairly concentrated in function space because each coefficient $\tanh(\theta_i)$ follows a highly bimodal distribution at $-1$ and $1$ as a result of the prior standard deviation being much larger compared to the scale at which $\tanh$ saturates (which is approximately 2).
4) Therefore, FS-MAP will better approximate the Bayesian model average than PS-MAP, following a similar argument in Section 4.2 line 271-274, and therefore achieve better generalization.

Correspondingly, there are situations where FS-MAP will likely underperform PS-MAP:

1) When the prior is poorly specified. For example, in a regression setting, if the observation noise is extremely over-estimated, FS-MAP may severely underfit the data because the log determinant regularization can overpower the likelihood. We demonstrate this effect in Appendix B.7, Figure 8.
2) When the log determinant contains singularities. As discussed in Section 4.1, in this case, FS-MAP leads to pathological solutions that ignore the data, though L-MAP mitigates this issue.

We hope this response addresses the potential concern that FS-MAP doesn't always outperform PS-MAP, and our thorough and transparent demonstration of the pros and cons of both approaches is viewed as a strength.

---

**Thank you for reviewing our work!**

---

### Decision · Program_Chairs · 2023-09-21

**Decision:**

Accept (poster)

**Comment:**

This paper studies learning most likely functions vs. parameters in the context of MAP estimation. Differences in parameterization affect the parameter space, and the authors provide an intellectual discussion on contrasting this approach to function space MAP. Reviewers have pointed out several dimensions of improvement to maximize clarity and I advise the authors to carefully address them in the camera-ready version. In particular, I find the limitation of the work to be inadequately discussed. For example, the algorithmic takeaway from the discussion in the paper is not clear as multiple reviewers have noted. This is fine within the scope of a single paper but is worth highlighting for future research.